# Nanoscale 3D DNA tracing in non-denatured cells resolves the Cohesin-dependent loop architecture of the genome in situ

K. S. Beckwith [1,2,11], Ø. Ødegård-Fougner[1,8,11], N. R. Morero [1], C. Barton [3,9], F. Schueder [4,5,10], W. Tang[6], S. Alexander [1], J- M. Peters [6], R. Jungmann [4,5], E. Birney [3] & J. Ellenberg [1,7] ✉

The spatial organization of the genome is essential for its functions, including gene expression and chromosome segregation. Phase separation and loop extrusion have been proposed to underlie compartments and topologically associating domains, however, whether the fold of genomic DNA inside the nucleus is consistent with such mechanisms has been difficult to establish in situ. Here, we present a 3D DNA-tracing workflow that resolves genome architecture in single structurally well-preserved cells with nanometre resolution. Our findings reveal that genomic DNA generally behaves as a flexible random coil at the 100-kb scale. At CTCF sites however, we find Cohesin-dependent loops in a subset of cells, in variable conformations from the kilobase to megabase scale. The 3D-folds we measured in hundreds of single cells allowed us to formulate a computational model that explains how sparse and dynamic loops in single cells underlie the appearance of compact topological domains measured in cell populations.

The three-dimensional organization of the human genome covers a large hierarchy of spatial scales, from individual nucleosomes of a few nanometres to supra-chromosomal compartments of several microns, and plays a central role in regulating genome functions[1]. Crosslinking and sequencing based chromosome conformation capture methods (e.g. 3 C/Hi-C) have defined compartments at the megabase (Mb) scale and topologically-associating domains (TADs) of several hundred kilobases (kb) as conserved features of genome organization[2,3]. While large scale compartments are believed to be driven by phase separation[4], TADs have been shown to fall between CCCTC sequence elements bound by CTCF[3] and depend on the Cohesin complex, a molecular motor that can extrude DNA loops in vitro[5–8]. Polymer simulations recapitulating HiC data have suggested that compact domains inside cells could thus arise from promiscuous loop extrusion in chromosomal DNA constrained by flanking CTCF sites[5,9–11]. Recent observations of the mobility of fluorescently labelled CTCF-sites in live cells are also consistent with loop extrusion by Cohesin bounded by CTCF[12,13].

To better understand structure-function relationships of the genome, directly resolving the nanoscale three-dimensional (3D) folding of genomic DNA (3D-fold) in single cells would be greatly beneficial. However, contact frequencies between genomic loci after crosslinking and sequencing as done in Hi-C do not directly measure the 3D-fold and typically average over millions of individual cells[3] and

[1]Cell Biology and Biophysics Unit, European Molecular Biology Laboratory, Heidelberg, Germany. [2]Dept. Biomedical Laboratory Science, Norwegian University of Science and Technology, Trondheim, Norway. [3]European Bioinformatics Institute, European Molecular Biology Laboratory, Hinxton, UK. [4]Faculty of Physics and Center for Nanoscience, Ludwig Maximilian University, Munich, Germany. [5]Max Planck Institute of Biochemistry, Martinsried, Germany. [6]Research Institute of Molecular Pathology, Vienna Biocenter, Vienna, Austria. [7]Karolinska Institutet, KTH Royal Technology College, Stockholm University, Stockholm, Sweden. [8]Present address: Department of Molecular Cell Biology, Institute for Cancer Research, The Norwegian Radium Hospital, Oslo, Norway. [9]Present address: Department of Computer Science and Information Systems, Birkbeck, University of London, London WC1E 7HX, UK. [10]Present address: Department of Biology, ETH Zurich, Zurich, Switzerland. [11]These authors contributed equally: K. S. Beckwith, Ø. Ødegård-Fougner. ✉e-mail: jan.ellenberg@embl.de

attempts to reconstruct higher order 3D-folds in cryo-preserved specimen by electron microscopy have failed due to the lack of sequence specific labels and the very high density of DNA in the nucleus[14]. By contrast, light microscopy-based methods such as the multiplexed labelling of short unique genomic sequence elements by oligonucleotide fluorescence in situ hybridization (FISH) probes in principle allow direct, single-cell visualization of genome structure[15–25]. Indeed, spatial genome imaging could already determine loop and TAD-scale structures correlating with HiC while revealing a strong cell-to-cell variability in genome folding[18–26]. So far however, comparatively harsh denaturing DNA-FISH hybridization conditions had to be used, that have previously been reported to potentially disrupt the fragile nanoscale structure genomic DNA[27–29]. Thus, a detailed structural characterization of nanoscale DNA architecture in individual, non-denatured cells is currently lacking.

In this study, we aimed to deploy the power of in situ genome imaging to investigate the nanoscale 3D-folding of single chromosomal DNA molecules down to individual loops in non-denatured human cells. We therefore developed a scalable high-resolution DNA tracing workflow that uses non-denaturing labelling with oligopaint FISH probes with a genomic resolution of better than 10 kb, and which provides improved structural preservation compared to denaturing FISH approaches. We combine this gentle, yet robust and precise labelling with automated sequential 3D imaging[30] giving a spatial precision of better than 30 nm in 3D. Using this approach, which we term "LoopTrace", we systematically investigate the 3D-folding of chromosomal DNA in multiple genomic regions from the kb to Mb-scale in single human cells in the presence and absence of Cohesin and its regulators. We show that genomic DNA folds like a random-coil polymer in the absence of Cohesin and is compacted at kilobase to megabase scales by Cohesin without CTCF. We measure the structure and variability in individual CTCF-anchored loops and the complex longer-range DNA interactions of entire topological domains at the megabase scale in 3D. The structural features we measure allow us to strongly constrain a computational model of genome folding with experimental data, leading to further insight into the properties of loop extrusion inside cells and explains how they give rise to both consistent and variable features in genome folding from cell to cell.

## Results

### LoopTrace enables nanoscale 3D DNA tracing with improved structure preservation

To enable precise and efficient resolution of the nanoscale 3D-fold of DNA in single cells, we deployed a DNA tracing method based on sequential imaging and centroid fitting of diffraction-limited regions tiled with barcoded OligoPaint probes, building on recent DNA tracing approaches[18,20,30] (Fig. 1a). In diploid hTERT-RPE-1 cells, we used 12-bp imaging probes targeting barcode sequences inspired by DNA-PAINT[31,32], allowing rapid 10 min exchange cycles and full 40-cycle (covering several genomic regions with position barcodes, region barcodes and controls) exchange experiments measuring up to 1000 cells in a 24 h experiment. We validated that our approach offered a high signal to background (mean = 7.8, Supplementary Fig. S1a) and fidelity (mean = 79%, Supplementary Fig. S1b) with a precision of better than 20 nm in each dimension ($x = 9 \pm 9$ nm, $y = 8 \pm 25$ nm, and $z = 18 \pm 35$ nm, median±std, Fig. S1c) without probe loss, for example during 30 exchange cycles of 300 kb regions on four chromosomes tiled with 10 kb probes, determined by re-labeling and re-imaging specific barcodes (Fig. 1a, Supplementary Fig. S1d), as well as high experiment-to-experiment reproducibility (Supplementary Fig. S1e, average Pearson's $r = 0.98$).

To investigate which FISH protocol would best preserve genome structure and yet allow the high labelling efficiency required for reliable DNA tracing, we compared a previously used denaturing FISH approach[18] that uses a comparatively short (3 min) heat treatment at 86 °C[18,21,24], with non-denaturing FISH based on enzymatic strand resection (CO-FISH/RASER-FISH, Fig. 2a)[28,33–35]. Qualitative evaluation showed that traces from denatured cells had dense clusters of the labelled loci spaced by larger spatial distances, while in non-denatured cells the loci were more homogeneously distributed along the trace, indicating differences in the preservation of DNA structure at this genomic scale (Fig. 2b). To quantitatively compare the two FISH approaches we derived structural metrics across four different 300 kb regions traced at 10 kb resolution (Fig. 2c–g, Supplementary Fig. S2). To allow comparison with available HiC data of the same regions, we computed median pairwise physical distances from our 3D traces (Fig. 2c, Supplementary Fig. S2a). While both FISH approaches correlated strongly with the HiC frequencies (Supplementary Fig. S2b, Pearson's $r = -0.93$ to $r = -0.96$) consistent with previous reports[18], direct comparison between them showed significant differences between denaturing and non-denaturing FISH. Consistent with the local clustering observed in single traces (Fig. 2b), denatured cells had higher variability in spatial distances between probes, leading to broad distributions of pairwise distances (Fig. 2d, e; Supplementary Fig. S2c, d) whose median was increased by about 30% (Fig. 2d, Supplementary Fig. S2c) and whose standard deviation was more than threefold higher than in non-denatured cells (Fig. 2f).

To understand if this increased heterogeneity in denatured cells was due to structural perturbation, we assessed the different steps in the denaturing and non-denaturing FISH procedures using two

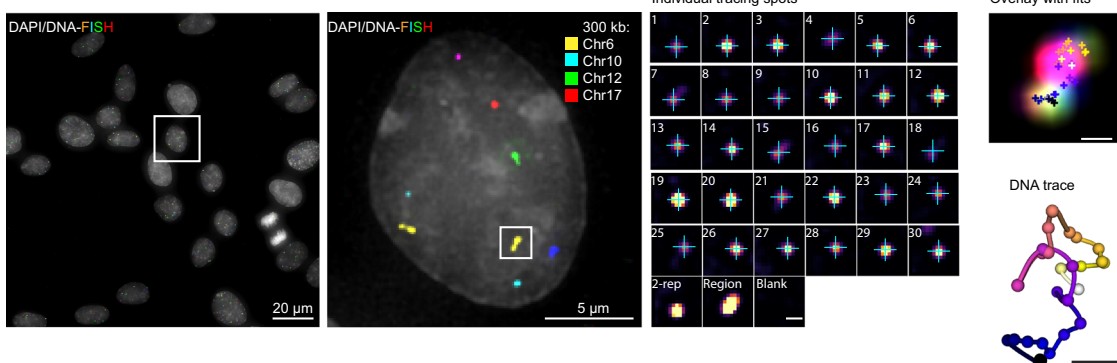

**Fig. 1 | LoopTrace enables high precision 3D DNA tracing in non-denatured cells.** Maximum projection of RPE-1 cells labelled with DAPI (grey) and oligoFISH probes (different genomic regions are indicated with pseudocolors), individual spots during 30 exchange cycles with centroid fits, projection of sequential imaging steps and centroid fits, and resulting 3D DNA trace. ID of individual tracing spots is equivalent to barcode ID in Supplementary Fig. S1a, b. Data representative of $n = 1561$ cells from 2 independent experiments. Scale bar in overlay 200 nm, for DNA-trace 100 nm.

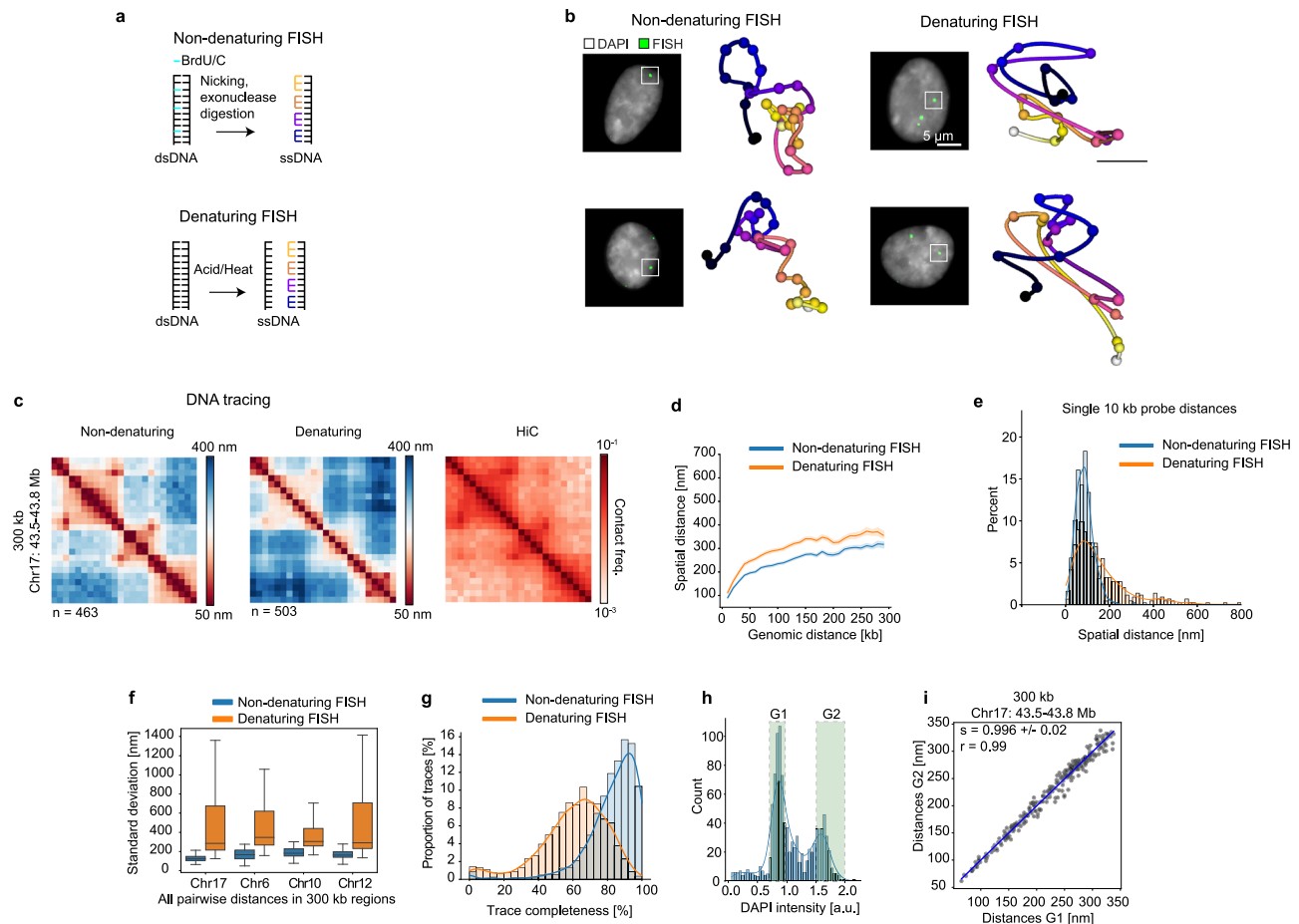

**Fig. 2 | Comparison of denaturing and non-denaturing FISH. a** Schematic of denaturing vs. non-denaturing FISH. **b** Two representative individual traces of 300 kb regions on chr 17 traced at 10 kb resolution with each FISH method in RPE-1 cells. Data representative of $n = 1271$ and $n = 1514$ cells from two independent experiments. Scale bar 100 nm. **c** Median pairwise distance maps of the same region as in b with denaturing (left column, $n = 463$ traces) and non-denaturing (middle column $n = 503$ traces). HiC contact frequencies (right column) from publicly available RPE-1 HiC data (GSE71831). Only cells assigned to G1 phase by DAPI intensity were used (-35%). **d** Spatial distance scaling of the same region as in (c) with the two FISH approaches. Shaded areas indicate 95 % confidence intervals around the median line. **e** Representative pairwise distance distributions between two consecutive 10 kb positions for the two FISH approaches. The histogram is overlayed by a kernel density estimate. Number of G1 traces analysed in (**c–e**) as indicated in (**c**). **f** Standard deviation of all pairwise distances from 300 kb regions

on four chromosomes with the two FISH approaches. Boxes indicate the lower and upper quartiles and whisker indicate the data range with 1.5 times interquartile range (IQR). Number of traces per condition as indicated in Supplementary Fig. S2a. **g** Distribution of trace-completeness measured by percentage of spots passing quality control thresholds for denaturing ($n = 2575$ traces), and non-denaturing ($n = 4505$ traces) conditions after removal of systematically under-performing barcodes. The histogram is overlayed by a kernel density estimate. **h** The integrated DAPI intensity in the central z-plane per nuclei was used to assign cells to G1 or G2. **i** Correlation between median pairwise distances in a 300 kb region from cells separated into G1 and G2 populations with the non-denaturing FISH approach. Linear regression (blue line) and the resulting slope (s) and Pearson's r are indicated. Data in (**b–i**) is representative of two independent experiments per condition. Source data are provided as a Source Data file.

complementary imaging approaches, the overall genomic DNA organization in the nucleus (using DAPI labelling) and the structure of TAD-sized replication domains[36] (using co-replicative fluorescent dUTP labelling) and visualised these throughout the FISH procedure by high resolution structured illumination microscopy (Supplementary Fig. S3)[37]. Our results show that acid and heat treatment perturbed the overall genomic DNA structure (Supplementary Fig. S3a-f, k), especially at the nuclear rim and the non-attached side of the nucleus in adherent cultured cells (Supplementary Fig. S3h-j). Furthermore, these denaturing treatments caused a shift in the relative positioning of replication domains (Supplementary Fig. S3a-f). These results are consistent with previous observations of the perturbations to genome structure introduced by DNA denaturation[27,29] and a recent direct comparison of denaturing FISH and RASER-FISH[28]. Taken together, these results suggest that denaturing FISH protocols perturb both global as well as local genome organization more strongly than non-denaturing protocols and thereby cause a substantial structural

variability in genome tracing, altering spatial scaling as well as pairwise distance distributions. These perturbations make accurate interpretation of single DNA traces, especially at the mechanistically important scale of individual loops much more challenging. By contrast, the non-denaturing approach shows less structural variability between cells and significantly improved tracing efficiency (Fig. 2g, 76% vs. 20% of traces over 80% complete). Further, due to the enzymatic digestion of the probe-targeted strand in one of the replicated sister chromatids inherent to the non-denaturing protocol[28,35,38], there is no issue of overlap between signals from closely adjacent sister chromatids in G2 cells[18,39], allowing a direct comparison (Fig. 2h, i). In four 300 kb regions, median pairwise distances between G1 and G2 cells were indistinguishable (Fig. 2i, Supplementary Fig. S2e), consistent with previous measurements by HiC of synchronized cells[5]. As the non-denaturing high resolution and high efficiency genome tracing approach we deploy here, performs well down to the level of individual loops[3], we term this approach "LoopTrace".

## LoopTrace visualizes the structure and heterogeneity of single loops between convergent CTCF sites directly in 3D

With an approach to measure the nanoscale 3D-fold in single, non-denatured cells, we proceeded to investigate the spatial features and cell-to-cell variability of a strong "loop domain" previously characterized by HiC[3]. To assess the roles of Cohesin and CTCF on DNA structure, we traced the same regions in RPE-1, HeLa wild type (WT) cells and HeLa cells with the Cohesin complex subunit RAD21 or CTCF homozygously tagged with an AID degron tag for acute depletion by auxin treatment (HeLa ΔRAD21 and ΔCTCF)[5]. We first targeted a region spanning a pair of occupied CTCF sites in convergent orientation[3] (previously mapped by Chip-seq) 100 kb apart near the *MEOX1* gene on chromosome 17 (Fig. 3a). Individual traces showed variable conformations in this region, including traces with large, intermediate or short distances between the two CTCF sites (Fig. 3b). To visualize both the average structure and its variability, we calculated a consensus representation of the region by general Procrustes analysis (see methods), highlighting the convergent CTCF sites and standard deviation of each position (Fig. 3c). In RPE-1 cells, the consensus representation showed a reduced distance and reduced standard deviation between the probes at the CTCF sites as the main conserved features. To systematically explore the conformational variability of this region, we calculated the Pearson distances between all individual traces, and ordered the traces by similarity through spectral seriation[40] (Fig. 3c, see methods). The resulting correlation maps showed highly divergent conformations with an average Pearson's distance of 0.93 (Supplementary Fig. S4a). Nevertheless, by visualising groups representing 1/6th of the spectrally ordered traces as consensus representations we observed a progressive change from an open fold via an intermediately compacted fold to a fully looped state (Fig. 3c), matching the observations from single traces (Fig. 3b) and indicating that individual RPE-1 cells adopt diverse conformations including both loop-like and more open states in the loop domain near the *MEOX1* gene. A similar result was obtained by seriation analysis of an 80 kb loop domain on chromosome 9, traced at an increased resolution of only 4 kb in RPE-1 cells (Supplementary Fig. S4b).

Interestingly, in HeLa cells the spectral seriation analysis of the genomic region close to *MEOX1* revealed differences in folding compared to the RPE-1 cells (Fig. 3d). Consensus representations from the seriation analysis highlighted additional interactions within the loop domain that caused a more closed fold to be adopted in all the seriation groups, although the proximity between the two CTCF sites varied between the groups (Fig. 3d) and the average Pearson distance was maintained at 0.93 (Supplementary Fig. S4a). Upon Cohesin depletion, the loop-like state and additional interactions were lost (Fig. 3e), consistent with the essential role of Cohesin in loop formation[5,6]. Spectral seriation analysis highlighted only minor variations of open conformations, and the average Pearson distance was reduced to 0.84, indicating more similar pairwise distances between different traces (Fig. 3e, Supplementary Fig. S4a). As a control, we performed seriation analysis of simulated ideal random coils, which resulted in consensus representations very similar to Cohesin-depleted cells (Supplementary Fig. S4c). Consensus representations of seriation groups from CTCF-depleted cells also indicate no fully looped state involving close proximity between the bounding CTCF sites, but continued to show states with increased compaction compared to Cohesin-depleted cells (Fig. 3f), consistent with CTCFs role as a boundary factor for loop formation[5,41]. These results demonstrate that spectral seriation-based clustering of single-cell traces is a useful tool to identify the diverse DNA folds underlying population-wide averages in an unbiased manner.

To better understand how the internal structural features of the loop domain near the *MEOX1* gene depend on Cohesin and CTCF, we compared the distributions of pairwise distances between specific probes located at the CTCF boundaries, and either inside or outside the loop domain (Fig. 3g). This analysis showed that Cohesin and CTCF most strongly alters the physical distance between the CTCF cites, while other positions in the loop domain showed only minor changes to their pairwise distance distributions in response to removing Cohesin or CTCF (Fig. 3g). To visualise and measure structural properties of the loop domains our tracing had caught in the "looped state", we used the stringent distance threshold of 100 nm (Supplementary Fig. S4d) between the two CTCF sites (Fig. 3h). Consistent with the seriation analysis, the looped states in RPE-1 and HeLa WT cells were more compact and yet were more elongated than the same region in Cohensin-depleted cells, while the total length of the trace remained unchanged (Fig. 3i). Thus, our high-resolution structural measurements indicate that CTCF and Cohesin cause compaction in the *MEOX1* loop region mainly by bringing loop anchors together, without measurable compaction of the internal fibre.

To determine the applicability of our seriation analysis to other Cohesin-dependent "loop domains" we repeated the analysis for a region on Chr12 spanned by the two convergent CTCF sites, which are spaced by 170 kb (Supplementary Fig. S4e), but which show a less clear corner-peak than the *MEOX1* region (Fig. 3a, Supplementary Fig. S4e). Intriguingly, in WT cells none of the seriation groups showed a clear loop forming between the CTCF sites, but rather highlighted compaction near the downstream CTCF site, intermediately compacted states and a looping state mainly involving the upstream CTCF site (Supplementary Fig. S4e). Visualising individual traces from each group supports the interpretation that diverse folds span this region without completely bridging the convergent CTCF sites (Supplementary Fig. S4e) explaining the lack of a clear corner-peak in the median distance maps.

Together, the unsupervised ordering of thousands of single-cell snapshots into structurally similar groups suggests that even strong loop domains[3] show a substantial cell-to-cell diversity and can adopt open, random-coil-like states, intermediately compacted or loop-like states in individual cells. The looped conformations clearly depend on CTCF and Cohesin, consistent with observations by HiC[5,6,41], DNA tracing[18,25] and live cell imaging[12,13]. While chromatin structure differences among different cell types have been observed previously[3,18], these results further highlight the ability of our high resolution single cell measurements and unsupervised computational analysis to identify differences in conformational variability at the level of individual loops between cell types that are not readily visible in averaged pairwise distance maps.

## Cohesin-dependent genome compaction at the 300 kb scale

To investigate how Cohesin and CTCF influenced compaction of genomic regions larger than single loop domains, we traced four 300 kb regions at 10 kb resolution (Fig. 4a, b; Supplementary Fig. S5a, b) and analysed the physical versus genomic distance scaling in 3D. Fitting power law scaling exponents this scaling analysis from the four regions (Fig. 4c, Supplementary Fig. S5c) showed that upon Cohesin depletion, 3D distances increased significantly, with scaling exponents increasing from $v_{control} = 0.37 \pm 0.09$ to $v_{\Delta RAD21} = 0.47 \pm 0.03$, scaling very close to an ideal random coil with $v = 0.5$. However, regional differences were observed. For example, the region on Chr6 was already unconstrained in WT cells and did not further decompact upon Cohesin depletion. In contrast to Cohesin depletion, the traced regions in cells depleted of CTCF remained overall similarly compact to control cells (Fig. 4c, d; Supplementary Fig. S5c, d; $v_{\Delta CTCF} = 0.34 \pm 0.03$ compared to $v_{control} = 0.37 \pm 0.09$), in line with expectation that genome scaling should reflect the abundance and length rather than the reproducible genomic positioning of loops[5,9,25]. Interestingly, CTCF depletion compacted the region on Chr6 to match the other regions (Supplementary Fig. S5b), suggesting that in this region CTCF prevents Cohesin from extruding longer loops, reminiscent of how CTCF acts to prevent looping across TAD boundaries[5,41].

 

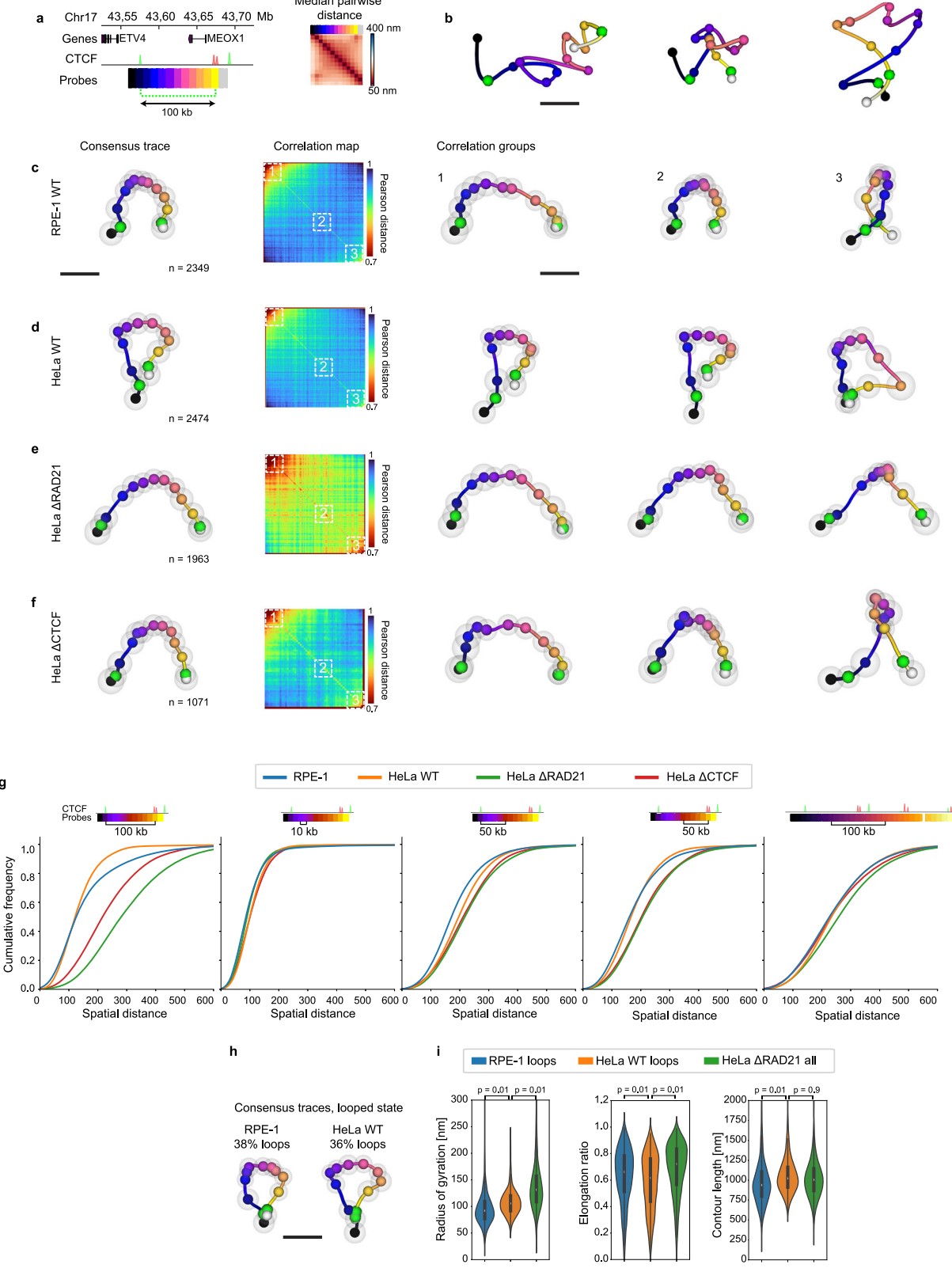

Interestingly, Cohesin depletion only showed a clear effect on physical scaling above about 50 kb (Fig. 4c, Supplementary Fig. S5c), which is consistent with HiC data also showing a similar contact frequency after Cohesin depletion for genomic distances below 50–100 kb[5,6]. To better understand the Cohesin-dependent folds that lead to compaction at genomic distances of 50–300 kb, we quantified the number of contacts and radius of gyration per trace below 100 nm

3D distance and above 50 kb genomic distance (Fig. 4d, Supplementary Fig. S5d, e). The four 300 kb regions had a median of four such contacts per trace in WT or CTCF-depleted cells which decreased to two contacts upon Cohesin depletion (Fig. 4d, Supplementary Fig. S5e). The median genomic length of these contacts was 100 kb (Supplementary Fig. S5f). At genomic distances of 200-300 kb, around half of the contacts below 100 nm were formed by stacking of smaller

**Fig. 3 | Structural features and variability of a single loop domain. a** Region targeted for tracing around the *MEOX1* gene on chromosome 17. Convergent CTCF-bound regions coloured green for (+) and red for (−) orientation (HeLa CTCF Chip-seq, ENCFF111RWV) predicted to form loops is indicated by the dashed green line. **b** Representative examples of individual traces from n = 1113 RPE-1 cells, with the position of CTCF sites indicated by green spheres. **c–f** Consensus representations, difference matrix $(1-\sqrt{PCC})$ between individual traces after sorting by the Fiedler vector and the consensus representations from each numbered seriation group from **c**, RPE-1, **d**, HeLa wild type (WT) cells, **e**, HeLa RAD21-mEGFP-AID or **f**, HeLa CTCF-mEGFP-AID cells depleted of Cohesin or CTCF for 2 h by auxin treatment. CTCF sites indicated by green spheres, grey spheres indicate the standard deviation of each position compared to the consensus representation. **g** Cumulative pairwise distance distributions between the indicated positions from traces shown in (**c–f**). **h** Consensus representations for traces filtered for a <100 nm distance between the

two CTCF sites for RPE-1 and HeLa WT cells, with the percentage of traces selected as indicated. **i**, Structural measurements of traces filtered as in **h** for the indicated cell types. Single-trace measurements of radius of gyration (HeLa ΔRAD21 = 131 ± 46 nm, HeLa WT = 106 ± 25 nm, RPE-1 = 91 ± 32 nm), contour length (HeLa ΔRAD21 = 1005 ± 368 nm, HeLa WT = 1022 ± 235 nm, RPE-1 = 939 ± 313 nm) and elongation (the ratio between major and minor axis of loci distribution, HeLa ΔRAD21 = 0.72 ± 0.19, HeLa WT = 0.61 ± 0.21, RPE-1 = 0.66 ± 0.20). Violin plots show median values, boxes indicate lower and upper quartiles and whiskers show data range within 1.5 times IQR. Indicated *p*-values calculated by non-parametric Conover's posthoc test with Bonferroni-Holm correction following a Kruskal-Wallis test for overall significance based on individual traces pooled from two experimental replicates. The number of traces for each experimental condition are indicated in (**c–f**). Scale bars 100 nm. Source data are provided as a Source Data file.

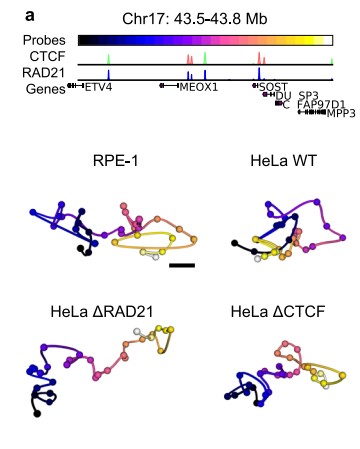

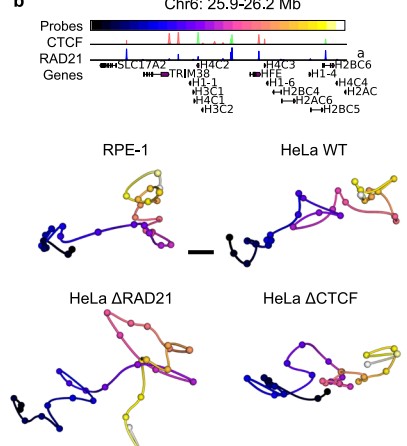

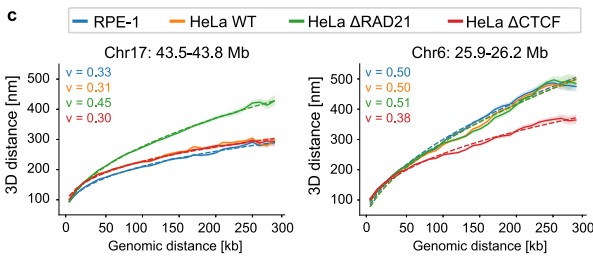

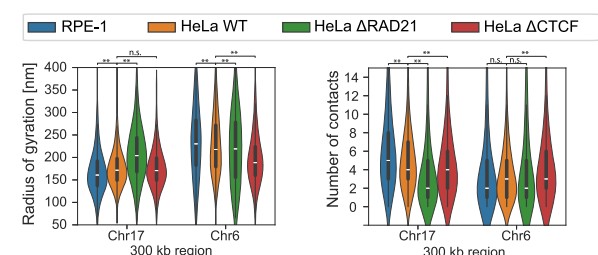

**Fig. 4 | Cohesin-dependent genome compaction at the 300 kb scale. a**, **b** Genomic overview and single trace examples of two 300 kb genomic regions traced at 10 kb resolution. CTCF ChIP-seq peaks are color-coded by orientation (green forward, red reverse). Scale bars 100 nm. **c** Median 3D distance scaling from two 300 kb genomic regions in the indicated cells lines. Scaling coefficients from a power law fit ($d_{spatial} = (d_{genomic})^{\nu}$) are labelled and the resulting fits shown as dashed lines. Shading indicates 95% confidence interval of the median. **d** Radius of

gyration and number of contacts <100 nm for the two regions in the indicated cell lines. See Supplementary Fig. S5 for detailed annotation of the number of traces per region and condition. Violin plots show median values, boxes indicate lower and upper quartiles and whiskers show data range within 1.5 times IQR. n.s.: not significant, *p < 0.05, **p < 0.01. Indicated *P*-values calculated by non-parametric Conover's posthoc test with Bonferroni-Holm correction following a Kruskal-Wallis test for overall significance. Source data are provided as a Source Data file.

loops (defined by two loops sharing an anchor to form a 3-way contact) (Supplementary Fig. S5g)[9,18,25]. Interestingly, CTCF-depleted cells had a similar proportion of stacked loops, suggesting that at this genomic scale there is a sufficient density of Cohesin to create stacked loops also in the absence of CTCF.

Taken together, the precise physical measurement of chromatin folds across four 300 kb regions in thousands of individual unperturbed or Cohesin- or CTCF-depleted cells shows that on average each 300 kb genomic region adopts unconstrained, random-coil-like folds in the absence of Cohesin, while in the presence of Cohesin the regions are constrained by two 100 kb Cohesin-dependent loops into a 300 nm-diameter volume, unless CTCF positioning prevents Cohesin from forming extended loops in the region.

## The 3D-fold of a TAD-scale genomic region reveals sparse and open loops

Given that we could capture the heterogeneity of the 3D-fold of DNA with high fidelity at the ~100−300 kb scale, we next explored if Loop-Trace could also be used to resolve the 3D-fold of larger, TAD-scale (1-2 Mb) genomic regions[3]. To this end, we traced four 1.3 to 1.8 Mb sized regions on three chromosomes with typical TAD features in publicly available HiC data[3] (Fig. 5a, Supplementary Fig. S6a). We labelled these regions in HeLa cells with 10-12 probe sets targeting 5−10 kb loci spaced 100−150 kb apart, directly overlapping CTCF sites identified by public HeLa Chip-seq data (green dashed line in Fig. 5a) as well as intermediate loci to gain information on the behaviour of predicted loop anchors as well as internal structure (Fig. 5a; Supplementary

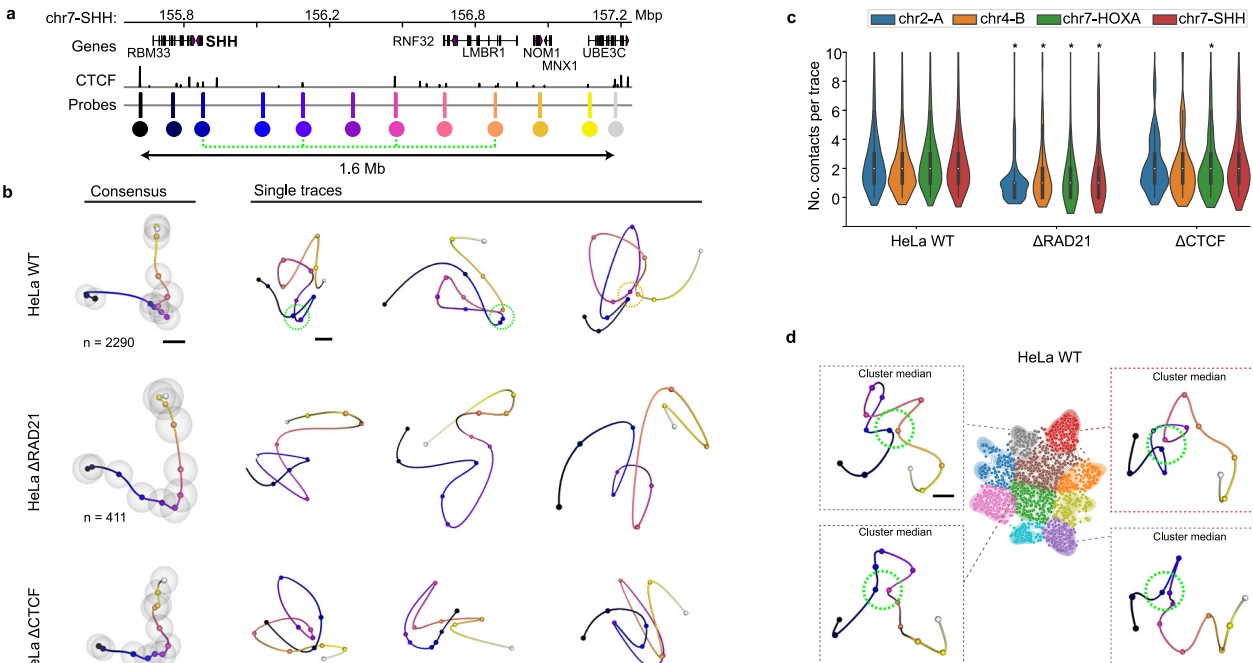

**Fig. 5 | DNA tracing in Mb-scale regions. a** Overview of a genomic region (Chr7-SHH) targeted with 12 probes (highlighted by coloured circles) across 1.6 Mb. Each probe targets a ~10 kb-sized region. Major CTCF sites (HeLa CTCF Chip-seq) predicted to form loops are indicated by the dashed green line. **b** Consensus representations and single trace examples from HeLa WT or HeLa RAD21-mEGFP-AID, CTCF-mEGFP-AID treated for 2 h with auxin. Data from 3 (WT) or 2 (AID cells) replicates, number of traces per condition are indicated. Green dashed circles indicate <100 nm proximity between probe no. 3 at the TAD boundary and downstream CTCF sites, while orange circle indicate three-way contacts (stacked loops). Grey spheres indicate standard deviation of each position compared to consensus representation. **c** Number of contacts (<100 nm) per trace for four 1.3-1.8 Mb regions on chr2, 4 and 7 in HeLa WT, ΔRAD21 and ΔCTCF cells. * $p < 0.05$ for

condition vs wild type, tested by Kruskal-Wallis non-parametric 1-way ANOVA followed by Conover's test with Holm adjustment for multiple comparisons. Violin plots show median values, boxes indicate quartiles and whiskers show data range within 1.5 times IQR. **d** UMAP with clustering of single traces by a Bayesian GMM (see methods) from the chr7-SHH locus in HeLa WT cells. Median distance reconstructions of exemplary clusters and insets highlighting the internal TAD architecture (3rd – 9th probe positions, dominating loops highlighted with green dashed ring) as indicated by trace/cluster colour. In total $n = 2290$ traces from 3 independent experiments, $n = 141$ (grey), $n = 174$ (red), $n = 365$ (brown) and $n = 273$ (purple) traces in the highlighted clusters. See Supplementary Fig. S6f for all cluster medians. Scale bars are 100 nm. Source data are provided as a Source Data file.

Fig. S6a). 3D DNA tracing of over 1000 single cells for each region revealed several close physical contacts consistent with the available HiC data (Supplementary Fig. S6a, b). Representing all data in 3D consensus representations revealed compaction inside the TAD regions as a conserved average structural feature across the four measured regions (Fig. 5b; Supplementary Fig. S6c, d). However, interactively exploring the fold of individual 3D traces revealed an overall rather open architecture of the megabase sized regions with, on average, only ~2 long-range physical contacts, typically between convergent CTCF sites (indicated by dashed green circles in single trace examples in Fig. 5b; quantification in Fig. 5c). The average contact number was significantly reduced to below 1 upon Cohesin depletion, while CTCF depletion did not affect the number of contacts, although they were repositioned within the regions (Fig. 5b, c; Supplementary Fig. S6c, d). With increasing genomic distance, we observed an increasing proportion of loops that coincided with two or more smaller loops, indicating that also at the Mb-scale smaller loops may stack to form multi-way interactions that bridge longer genomic stretches (Supplementary Fig. S6e; example in Fig. 5b indicated by orange circle), as predicted by loop extrusion simulations[9] and consistent with DNA tracing data from TAD-scale regions in single human[18] or mouse[25] cells.

To mine the wealth of TAD-scale DNA 3D-folds from thousands of single cells in an unbiased manner, we next performed unsupervised clustering of all single traces using a Bayesian Gaussian mixture model (see methods) to group these structurally more complex 3D-folds by similar structural features, using the *SHH* locus on chromosome 7 as an example (Fig. 5d; Supplementary Fig. S6f). Interestingly, traces

reconstructed from the median pairwise distances in each cluster revealed that most automatically identified groups represented a specific physical proximity between the upstream TAD boundary of the *SHH* gene and a different downstream CTCF site (Fig. 5a, d; all groups shown in Supplementary Fig. S6f). In these clusters, the presence of these loops brought the boundary in close proximity with a loop anchor inside the TAD, while the lack of loops across the boundary to outside the TAD led to comparatively increased physical distances to these regions (Supplementary Fig. S6f). These results provide a structural explanation for how a single strong anchor on one side of a loop can generate an effective boundary and yet lead to many different loop sizes and internal folds. Together, these highly variable and open folds will average to a more compact domain at the population level (Fig. 5b).

Overall, the 3D in situ structural data LoopTrace allowed us to generate at the larger scale of megabase sized genomic regions are fully consistent with our observations at the smaller, ~100-300 kb, scale of single DNA loops and domains. We can conclude that a typical TAD-scale region on average exhibits only sparse long-range looping interactions inside a single cell, that these loops are rather open and of variable size but can be stacked to generate physical proximity between genomically distant loci.

## A data-constrained polymer model predicts that loops and TADs are formed by a minor fraction of Cohesin complexes

Our high-fidelity measurements of 3D-folding of chromatin across thousands of cells should provide valuable experimental constraints for current loop extrusion models of Cohesin function[9,10]. To explore

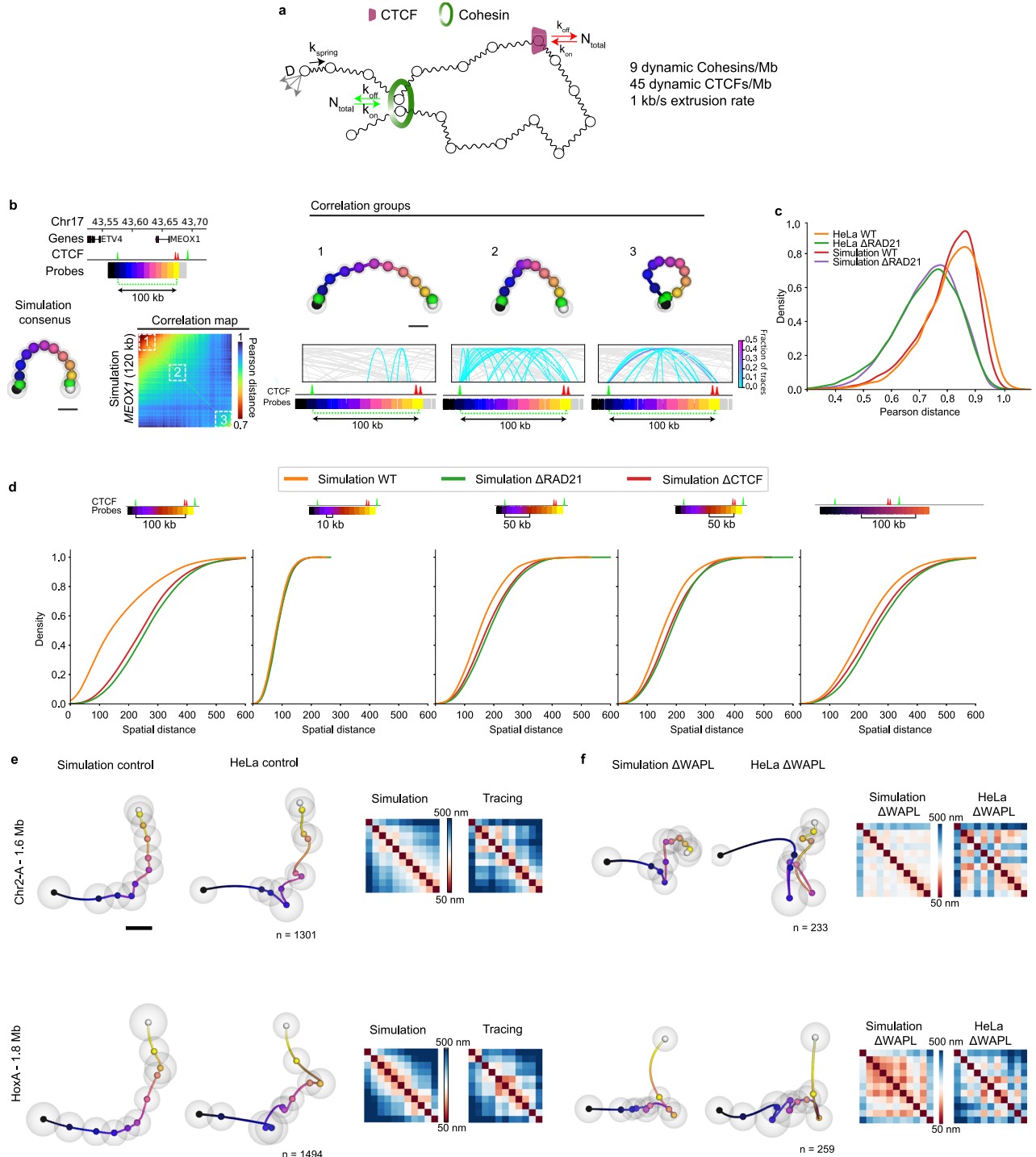

**Fig. 6 | Data-constrained computational model of DNA loops and TADs.**
**a** Schematic of the Rouse polymer/loop extrusion model which includes physiological values of the dynamic properties of chromatin, Cohesin and CTCF and is further constrained by 3D DNA tracing data. Best-fit parameters are displayed (see also Supplementary Fig S7). **b** Genomic overview, 3D consensus representation, difference matrix as well as consensus representations from the labelled seriation groups from the simulated 100 kb region at Chr17:43.6-43.7 Mb. Examples of ground truth loops from the simulation are shown with grey lines indicating loops with one or two anchors outside the displayed region, while the blue lines indicate loops inside the region. The thickness and colour of the lines scales by their frequency in the simulations. **c** Pairwise Pearson's correlations for individual experimental and simulated traces from Chr17: 43.6-43.7 Mb. Experimental data from two

independent replicates per condition. **d** Cumulative pairwise distance distributions between the indicated genomic positions in Chr17: 43.6-43.7 Mb for the indicated simulation conditions. **e** Simulated and experimental consensus representations and median pairwise distance maps from a 1.6 Mb region on Chr2 and a 1.8 Mb region on Chr7 traced with 11 probes with ~150 kb spacing (see Supplementary Fig. S5) in control cells and corresponding simulations. Grey spheres indicate standard deviation of each position compared to consensus representation. **f** The same genomic regions as shown in (**e**) after WAPL was depleted by auxin treatment of HeLa WAPL-Halo-AID cells for 2 h (HeLa ΔWAPL), while in simulations Cohesin residence time was increased 20-fold. Number of traces as indicated in (**e**) and (**f**) from three (HeLa control) or two (HeLa ΔWAPL) independent experiments. Scale bars in (**b**) and (**e**) are 100 nm. Source data are provided as a Source Data file.

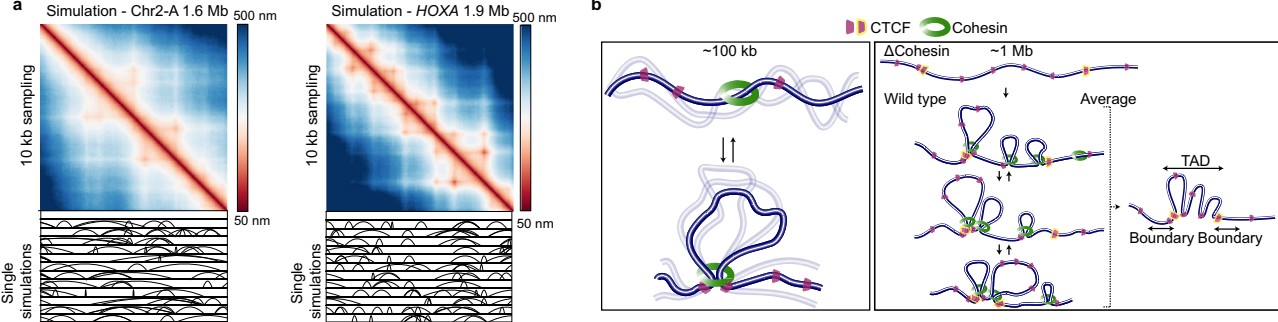

**Fig. 7 | Experimentally constrained loop extrusions simulations suggest how single loops and larger domains may be composed in single cells. a** Simulations of 1.6 Mb region on Chr2 and a 1.9 Mb region on Chr7 (matching experimental data in Supplementary Fig. S5), with exemplary ground truth loops (black arcs indicate the genomic positions bridged by the loops) from a single simulation run. **b** Proposed experimentally parametrised loop extrusion model for chromatin architecture at the scale of 100 kb and 1 Mb.

this further, we adapted polymer/loop extrusion simulations that have been used to model genome structure and constrained these with our measurements and complementary experimental data[9,10]. As input parameters we used an experimentally measured diffusion coefficient of chromatin[42] and set the spring constant of the Rouse chain to match our measurements of chromatin in Cohesin-depleted cells across the four high resolution 300 kb regions (Supplementary Fig. S7a, $r = 0.96$). For the loop extrusion simulations we constrained the parameters of our model systematically with the quantitative experimental data currently available for Cohesin complexes in single HeLa cells. Our model includes the experimentally determined cellular protein concentrations[43], dynamic chromosome residence times[43–45] as well as number and orientation of CTCF binding sites measured by Chip-seq in the HeLa cell genome[3,43] (see Supplementary Note. 1). For simulation purposes the parameters were adjusted to per-Mb values to be independent of cell ploidy. The main parameters we could not constrain was what fraction of the about 250 000 Cohesin complexes in a HeLa cell is actively extruding loops, and their in situ extrusion rate, which have been estimated to range from 0.1-1 kb/s[6,8,13].

To estimate these parameters, we compared simulations without CTCF to experimental 3D DNA traces from HeLa cells depleted of CTCF. The model reached the lowest root mean squared deviation (RMSD) compared to the single cell structural data from the 300 kb and Mb-scale regions when the fraction of active Cohesins was set to 30% (9 dynamically bound Cohesins/Mb) and the extrusion rate was 1 kb/s (Fig. 6a, Supplementary Fig. S7b). Finally, we added dynamically binding CTCF to the simulations, and observed the best correlation when CTCF concentration was set to the upper bound of the measured values (45 dynamically bound CTCFs/Mb, $r = 0.89$, Supplementary Fig. S7c)[43]. With these experimentally constrained parameters, we asked if we could replicate the pairwise distance distributions and variability measured in the loop domain near the *MEOX1* gene on chromosome 17 (see Fig. 4). Indeed, seriation analysis produced similar consensus representations as observed in RPE-1 cells (Fig. 6b), including open, partially compacted and fully looping states (Fig. 6b, d) as well as closely recapitulating the conformational variability between traces (Fig. 6c). The more consistently looped state observed in HeLa cells (see Fig. 3d) was not recapitulated based on CTCF-Cohesin interactions alone suggesting additional CTCF-independent interactions in this genomic region in HeLa cells. Inspection of the underlying progressively extruded loops from simulations showed that the most open seriation conformation had few internal loops, the partially compacted state had a mixture of partially and fully extruded loops bridging the CTCF sites, and the fully looped group was enriched in single Cohesin loops bridging the two CTCF sites (Fig. 6b). Overall, the loop extrusion simulations (Fig. 6b) predicted that between the

two 100 kb spaced convergent CTCF sites on chromosome 17, 18% of the traces contain a partially extruded loop while 35% contain a fully extruded loop, close to our estimate of 32% of Cohesin-specific loops between the two convergent CTCF sites (see Supplementary Fig. S4d).

### As predicted by the model, increasing Cohesin binding to DNA further extends loops

Given the ability of the model to predict the conformational variability and contact frequency of our experimentally observed 3D folds at the loop-scale, we asked the model to make a testable prediction for larger regions. As it is experimentally difficult to control the fraction of active Cohesin, we used the model to predict what structures would form if we forced active Cohesin complexes to bind 20 times longer to DNA, a situation that can be experimentally generated by removing its regulator WAPL for 2 h[46]. In this in silico scenario, the model predicted an increased average length of non-stacked loops from $393 \pm 14$ to $455 \pm 27$ kb and an increased proportion of stacked loops from $12 \pm 3\%$ to $20 \pm 8\%$ in the four Mb-scale regions (Supplementary Fig. S7d).

To validate this prediction experimentally, we acutely depleted WAPL from HeLa cells where the gene had been homozygously tagged with an AID degron tag for 2 h and then retraced the 3D-fold of this domain in over 300 individual cells. In striking agreement with the model prediction, we saw an increase in size and stacking of loops (Fig. 6e, f) that matched the simulated values with an error of <10% (Supplementary Fig. S7d), suggesting that the parameters we use in the model are able to predict the 3D-folding of genomic DNA with high accuracy up to the scale of 2 Mb.

Having validated the model in this way placed us in a position to predict the key parameters of individual loops and TADs assuming the physiological situation, where all Cohesin regulators including WAPL are present (Fig. 7). Here, our model predicts that in an average TAD-scale domain of around 1 Mb, between three and four Cohesins interact with a dynamically exchanging pool of on average about 45 CTCF molecules to generate consistent loop contacts between the dominant CTCF boundary sites and internal anchors up to genomic distances of ~300 kb. For a 100 kb single loop domain, the model would predict that one Cohesin interacts with four dynamically exchanging CTCFs leading to a single loop contact between convergent CTCF sites in about 30% of cells (Fig. 3h, Supplementary Fig. S4d). The simulation further predicted that across Megabase-scale domains multiple Cohesin loops compound to form long-range interactions (Fig. 7a), providing an explanation for the stacked loops we found in our traces.

## Discussion

The "LoopTrace" workflow we introduce here provides a precise, robust and scalable approach to investigate the nanoscale folding

principles of the genome in situ in structurally well-preserved nuclei of single human cells. Our method does not suffer from the increased structural variability we demonstrated to be a result of denaturation-based FISH methods and shows good structural conservation of TAD-sized replication domains and overall nuclear features of genomic DNA. A necessary step in the CO-FISH/RASER-FISH protocol is replicative labelling with BrdU/C, limiting the method to replicating cells in vitro or in vivo[28,38]. Nevertheless, BrdU and other nucleotide analogues have seen a number of applications in genomics, including strand-specific sequencing[47], replication timing[48] and sister-chromatid sensitive HiC[49]. It should be noted that BrdU can lengthen the cell cycle and induce genotoxic effects, especially at high (>50 μM) concentrations and over multiple cell divisions[50], so concentration and treatment duration should be adapted to the model system depending on the cell cycle length and BrdU sensitivity[48].

Our 3D DNA tracing with up to 30 nm/4 kb resolution provides direct, in situ measured structural data on the 3D-fold of the genomic DNA, which is key to understanding structure-function relationships of the genome. Based on HiC data and computer simulations[6,9,10], and live cell imaging of genomic loci[12,13], Cohesin mediated progressive loop extrusion had been proposed to be the mechanism that drives TAD formation. However, a comprehensive structural analysis of the resulting looping conformations of genomic DNA in individual, non-denatured human cells has so far been lacking. Our 3D tracing data in single cells shows that in the absence of Cohesin, genomic DNA at the scale of loops and TADs folds like a weakly interacting random coil polymer. Onto this rather open and dynamic structure, individual loops are superimposed, driving compaction that depends on Cohesin activity and are positioned at specific genomic sites dependent on CTCF, consistent with current interpretations of HiC and DNA tracing data[6,9,10,25,51]. Furthermore, our quantitative structural data allowed us to construct a fully parameterized computer simulation that predicts that single cells only use about 30% of all available Cohesin complexes for extrusion of loops larger than ~10 kb.

We find that by careful analysis of a strong 100 kb loop domain on chromosome 17 a Cohesin-dependent loop was present in about 30% of cells, broadly consistent with estimates from single-cell HiC[52] and live cell imaging[12,13], although estimates vary substantially based on the region studied and possible cell type differences. Our model predicts that when present, the loops were formed by a single Cohesin complex residing mainly in the fully extruded state, with a lower percentage of intermediately extruded loops. These simulations thus can recapitulate not only average structures, but also the increased relative variability at loop anchors and the conformational variability between individual cells. However, how loop domains are structured varied with genomic region. For example in a loop domain on chromosome 12, our analysis revealed that this domain did not form by single loops, but rather different combinations of smaller loops and folds within the domain, which was also the case for larger, TAD-scale domains, consistent with recent DNA tracing data in mESCs[25]. Thus, the complete set of TAD features, i.e. strong boundaries and high frequency of internal interactions did not exist in any single cell at any time point. Rather, TAD boundaries arise from the cumulative effect of dynamically positioned sparse looping interactions inside the extrusion range of strong CTCF sites in cell population data.

Our simulations included dynamic exchange of CTCF and Cohesin, consistent with experimental observations[44,45,53], and estimate a Cohesin extrusion rate of 1 kb/s, which is in the range of current estimates from HiC[6] and live cell imaging[13]. However, consolidating our simulations with the experimentally determined loop and TAD structures indicated that a high CTCF to Cohesin ratio (45 CTCF and 9 Cohesin dynamically bound per megabase) was required, which deviates from average experimental abundances in HeLa cells (23 CTCF and 32 Cohesins per Mb)[43]. For Cohesin, these findings suggest that a only a minor (30%) fraction of Cohesin complexes are involved in loop formation at a given time.

The increased CTCF abundance suggested by the simulations may be due to the assumption that all dynamically bound CTCFs have an equal chance of stalling a passing Cohesin, and that the occupancy of CTCF scales with ChIP-seq signal strength. Recent studies suggest that the looping frequency of specific loops and thus the resulting pairwise distance maps of the genomic regions may be more complex[54–56]. However, the mechanistic basis for this discrepancy remains unclear, and before making more complex assumptions in our simulations, such as constrained diffusion of CTCF and CTCF clustering[57] and/or stabilization of chromatin-bound CTCF by Cohesin[53], further investigations would be needed.

In conclusion, we anticipate that the LoopTrace approach we developed here, which combines oligopaint FISH with improved preservation of genomic structure, scalable high-precision 3D DNA tracing and computational approaches to interpret variability and consistent features between individual cells, will be a valuable tool to directly investigate the structure-function relationship of the genome at the nanoscale in single cells, and inform future mechanistic models. Due to labelling of only one sister chromatid, LoopTrace is also particularly well suited to studying the in situ structure of chromatin involved in replication, homologous DNA repair and cell division[53,58]. By increased multiplexing, DNA tracing is in principle scalable to whole chromosomes or genomes[22,59], and in the future should enable a deeper understanding of the link between genome architecture and the functional state of individual cells in health and disease.

## Methods

### Generation of HeLa cell lines

All AID-tagged HeLa cell lines used in this study were generated by homology-directed repair using CRISPR Cas9 (D10A) paired nickase[60]. HeLa-SCC1-mEGFP-AID and HeLa-CTCF-mEGFP-AID were described before[5]. Based on the cell line SCC1-GFP[5], we introduced a Halo-AID tag to the N-terminus of WAPL, generating Halo-AID-WAPL/SCC1-GFP. Subsequently, Tir1 expression was introduced by transducing a homozygous cell clone with lentiviruses using pRRL containing the constitutive promotor from spleen focus forming virus (SFFV) followed by Oryza sativa Tir1-3xMyc-T2A-Puro[5]. The gRNAs sequences used were: CACCGCTAAGGGTAGTCCGTTTGT and CACCGTGGGGA-GAGACCACATTTA. The primers used for genotyping were: TGATT TTTCATTCCTTAGGCCCTTG and TACAAGTTGATACTGGCCCCAA.

### Cell culture

RPE-1 cells (hTERT-immortalized retinal pigment epithelial cell line, ATCC No. CRL-4000, RRID: CVCL_4388) were grown in Dulbecco's Modified Eagle Medium (DMEM/ F-12, Cat. No. 11320074, Thermo-Fisher) supplemented with 10% FBS (Cat.# 26140, ThermoFisher) and 1% Gibco Antibiotic-Antimycotic (Cat.# 15240096, ThermoFisher). HeLa Kyoto (HK) cells (RRID: CVCL_1922) were a kind gift from Pr Narumiya, Kyoto University. HK WT, HK RAD21-mEGFP-AID (+OsTir1), HK CTCF-mEGFP-AID (+OsTir1) and HK RAD21-mEGFP Halo-AID-WAPL (+OsTir1) were cultured in high glucose DMEM (Cat.# 41965039, ThermoFisher) supplemented with 10% FBS, 1 mM sodium pyruvate (Cat.# 11360070, ThermoFisher), 2 mM L-glutamine (Cat.# G7513, Sigma) and 100 U/ml penicillin-streptomycin (Cat.# 15140122, ThermoFisher). Cells were incubated at 37 °C, 5% $CO_2$ in a humidified incubator. At 70-80% confluence cells were trypsinized with 0.05% Trypsin/EDTA (Cat.# 25300054, ThermoFisher) and transferred to a new culture dish at appropriate dilutions every 2–3 days. For micro-slide seeding (μ-Slide VI 0.5 Glass Bottom, Cat.# 80607, Ibidi), trypsinized cells were counted and diluted to ~5 × 10⁵ cells/ml in complete culture medium. For some experiments including AID-tagged cell lines, cell suspensions of each of the cell lines were separately labeled with either one or both of ViaFluor 488 SE and ViaFluor 405 SE

fluorescent dyes (Biotium) according to the manufacturer's instructions. Briefly, cell suspensions in 500 μL of PBS were mixed with the dye (1 μM) for 15 min at 37 °C and quenched by addition of 500 μL of cell growth medium and incubation for 5 min. Cells were centrifuged and resuspended in 500 μL of culture medium, incubated for 25 min at 37 °C and finally centrifuged again and resuspended in fresh medium at $5 \times 10^5$ cells/ml. Labeled cells were mixed in equal amounts, seeded in Ibidi dishes and grown for 24 h in the presence of 40 μM BrdU: BrdC mix (3:1) and 200 uM auxinole (Hycultec) to inhibit background degradation[61]. To deplete AID-tagged proteins, auxinole was washed out 22 h after cell seeding, and replaced with 500 μM Inole-3-acetic acid (IAA, Sigma) for 2 h. Degradation of AID-tagged proteins in HK cells under these conditions was previously determined[5], and verified again in this work by incubating HK RAD21-mEGFP-AID cells with IAA and/or auxinole for different times (from 15 min to 4 h) and quantifying the amount of target protein left by capillary electrophoresis (Jess Simple Western, ProteinSimple) (Supplementary Fig. S8a). Protein normalization was achieved by determining the total amount of protein loaded in each capillary with a fluorescent dye that binds to all amino groups in proteins (PN reagent, ProteinSimple).

## Primary FISH probe design

Unique FISH probes were designed using the human reference genome GRCh38.p12. Sequences used are listed in Supplementary Data 1. For the probes targeting regions on chromosomes 2, 4 and 6, 7, 9, 12 and 17, we used OligoMiner[62] with genomic target length between 36 and 42 nt, $T_m$ between 42 and 50 °C (accounting for 2XSSC with 50% formamide), and a minimum of 2 nt spacing. Selected probes were filtered by the 42 °C LDA model with 0.8 stringency and a maximum of 5 or 10 off-target 18 bp kmers. Up to 100 probes in 10 kb regions were used per position. Docking sequences for secondary imagers (two identical barcodes for each position and one for each region per oligo) and amplification primers were appended, and probe libraries were ordered from Genscript (Precise Synthetic Oligo Pools) and amplified (see below).

## Imager and primer sequence design and labeling

12-bp imagers targeting the probes for regions on chromosomes 2, 4, 6, 7, 9, 12 (named Dp1-38, 101-114) were sourced from a list of 12 bp human nullomers[32], filtered for low secondary structure (ΔG > −0.2 kcal/mol using Primer3[63]), $T_m$ < 40 °C (at ion concentration 390 mM), GC-content 40-50%, less than 3 consecutive identical nucleotides, and few high temperature (> 28 °C at 390 mM) heterodimers in the set. 11 and 10 bp off-target sites for the 12 bp probes were detected by BLAST + (blastn-short, E-value < 10000), and the probes were sorted ascendingly for 11 bp off-targets. 12 bp oligo imagers were ordered with 3' or 5'-azide functionalities (Metabion), reacted with Atto647N-alkyne or Atto643-alkyne (Attotec) using click chemistry (ClickTech Oligo Link Kit, Baseclick GmbH) according to the manufacturer's instructions with 100 μM oligo and 1.2-fold excess dye. Labeled oligos were purified by diluting to 20 μM in TE buffer, then adding 5-fold excess water-saturated n-butanol (Sigma), vortexing for 10 s, centrifuging (14 000 g, 1 min), extracted once more, and carefully pipetted from underneath the n-butanol layer into a fresh tube.

PCR primer sequences were selected from 20-mer sub-sequences of orthogonal 25-mer sequences[64], filtered for 55 °C > $T_m$ > 60 °C (at 50 mM salt, primer3), less than 4 identical nucleotide repeats, and starting and ending with a C/G. The primer sequences were further screened for compatibility with the 12 mer docking sequences by ensuring heterodimer $T_m$ under 15 °C (primer3, 390 mM ions) and under 40 °C hairpin $T_m$ (primer3, 390 mM ions) of sequences composed by combining all possible pairs of 10 000 filtered 20 mers with the T7 in vitro transcription promoter and 200 12 mers selected above. Forward and reverse primers were paired and sorted by the number of

12-mers the set was compatible with. All oligo sequences used in this work are listed in Supplementary Data 1.

## Amplification of FISH Libraries

The FISH library targeting chromosomes 2, 4, 6, 7, 9 and 12 was amplified from synthetic oligonucleotide pools (Genscript), through PCR amplification followed by in vitro transcription, reverse transcription and ssDNA purification, as described before[30,65].

Briefly, the PCR amplification protocol for each library and primer pair was optimized by monitoring the progression of the reaction in real time with a qPCR machine to observe the minimum number of PCR cycles necessary to reach the amplification plateau. PCR was performed using 75 μl 2x Phusion High-Fidelity PCR Master Mix (ThermoFisher), 15 ng oligonucleotide pool, 0.5 μM of each primer and water up to 150 μl. The following PCR protocol was used: (i) 98 °C for 3 min, (ii) 98 °C for 10 s, (iii) 66 °C for 10 s, (iv) 72 °C for 15 s. Steps (ii) to (iv) were cycled until the reaction approached its amplification plateau (14-18 cycles). The amplified dsDNA template was then purified with a DNA Clean & Concentrator-25 kit (Zymo Research) and eluted in 50 μl of water. 5 μl of this sample was set-aside for quality control by gel electrophoresis. In vitro transcription was done using the HiScribe T7 Quick High Yield RNA Synthesis Kit (NEB), following the manufacturer's instructions for short transcripts. Reaction was set up as follows: 1.5 μg DNA template, 10 mM NTP buffer mix, 6.25 μl T7 RNA Polymerase mix, 250 U RNAsin Plus RNAse inhibitor (Promega) and water to 160 μl. Amplification proceeded for 16 h at 37 °C. At this point, 1 μl of this sample was set-aside for quality control. ssDNA amplification from template RNA was performed by reverse transcription, using: 150 μl of unpurified in vitro transcription reaction, 1.7 mM dNTP mix, 19 μM forward primer, 240 U RNAsin Plus RNAse inhibitor (Promega), 1200 U Maxima H Minus Reverse Transcriptase (ThermoFisher), 60 μl of 5x Maxima Buffer and water up to 300 μl. The reaction mix was incubated for 1 h in a water bath at 50 °C. Template RNA was subsequently degraded by addition of 150 μl 0.5 M EDTA and 150 μl 1 N NaOH, and incubation for 15 min in a water bath at 95 °C. Amplified ssDNA library was purified with the HiScribe T7 Quick High Yield RNA Synthesis Kit (NEB). For this, the reaction solution (600 μl) was mixed with 1.2 ml Oligo binding buffer and 4.8 ml 100% ethanol, and then loaded into 2 × 100-μg capacity purification columns. Columns were washed twice with 250 μl washing buffer and ssDNA was eluted with 100 μl nuclease-free water per column. Library concentration was measured by spectrophotometry in a Nanodrop. Finally, to assess the specificity and efficiency of the amplification reaction, the intermediate products of each amplification step (PCR-amplified dsDNA, RNA and final ssDNA) were analyzed by Gel electrophoresis in denaturing conditions (Novex TBE-Urea Gels, 15%, ThermoFisher).

## Non-denaturing FISH

Non-denaturing FISH (RASER-FISH) was adapted from previously described protocols[28,33,38] to be performed in 6-channel microslides (μ-Slide VI 0.5 Glass Bottom, Ibidi), while minimizing DNA denaturation and genome structure perturbations and optimizing FISH signal for DNA tracing. Briefly, cells were seeded in microslide channels at a final concentration of ~5 × 10^5 cells/ml in 120 μL cell culture media containing 40 μM BrdU:BrdC mix (3:1, Cat.# B5002, Sigma and Cat.# 284555, Santa Cruz Biotec) and grown for 17−24 h. Afterwards, cells were washed once with PBS and fixed with 4% PFA (v/v, Cat.# 15710, EMS) in PBS for 15 min. Free aldehyde groups were quenched with 100 mM NH4Cl (Cat.# 213330, Sigma) in PBS for 10 min and cells were permeabilised in 0.5% Triton X-100 (v/v, Cat.# T8787, Sigma) in PBS for 20 min. To further sensitize BrdU/C-labelled DNA to UV light, cells were treated with 0.5 μg/ml DAPI (Cat.# D9542, Sigma) in PBS for 15 min and washed once in PBS. Microslides were exposed to 254 nm UV light for 15 min in a Stratalinker 2400 UV crosslinker, and cells were finally treated with 1U/μL Exonuclease III (Cat.# M0206, NEB) in 1X

NEBuffer 1 at 37 °C for 15 min in a humid container. All incubations were performed at room temperature, unless stated differently. For primary probe hybridization, cells were first incubated at 37 °C for 1 h in buffer H1 (50% formamide (FA, Cat. # AM9342, ThermoFisher), 10% (w/v) dextran sulfate (D8906, Sigma) (Cat.# R6513, Sigma) in 2xSSC (Cat.# AM9763, ThermoFisher) and afterwards with primary probes (~2 nM per primary probe) diluted in H1. Primary probes were hybridized overnight at 42 °C in a humid chamber). After primary hybridization, cells were washed twice in 50% FA in 2xSSC buffer at 35 °C (7–10 °C below probe on-target $T_m$) for 5 min, followed by 3 washes with 2xSSC containing 0.2% (v/v) Tween-20 (Cat.# P9416, Sigma). For the probe library targeting regions on chromosomes 2, 4, 6, 7, 9, 12 and 17, RNAse A was not included in the hybridization buffer and was replaced by an RNAse H (NEB) treatment (1:100 in RNAse H reaction buffer for 20 min at 37 °C) after primary hybridization to reduce non-specific background. For non-sequential secondary probe hybridization, fluorescently labelled 20-mer imagers were diluted to a final concentration of 20 nM in buffer H2 (25% FA, 10% (w/v) dextran sulfate, 0.1% (v/v) Tween-20, 2xSSC buffer) and incubated for 2 h at 30 °C. Washing steps were then conducted with 25% FA in 2xSSC buffer for 5 min, followed by washes with 2xSSC containing 0.2% (v/v) Tween-20. Before imaging, DNA was stained with 0.2 µg/ml DAPI for 10 min.

### Classical FISH

Standard FISH protocols including heat denaturation of DNA were performed in parallel to compare with the non-denaturing FISH procedures. We note that our step-by-step comparison by structured illumination microscopy used an increased detergent concentration and HCl incubation time compared to the some protocols, which may cause increased protein denaturation. RPE-1 cells were seeded in microslides, fixed and permeabilized as above and treated with 0.1 M HCl (Cat.# 320331, Sigma) for 5 min (15 min in step-by-step comparison). For primary probe hybridization, cells were incubated for 1 h in buffer H1 at 37 °C, followed by 1 h incubation in the same buffer containing the primary probes and a 3 min heat incubation step in a ThermoBrite slide hybridization system (Leica Biosystems), either at 75 °C or 86 °C. The temperature during incubation was verified with an additional external digital thermometer. Hybridization was then continued overnight at 42 °C. Washes and secondary hybridization steps were as described for the non-denaturing FISH protocol.

### Pulse DNA labelling with fluorescent nucleotides

Fluorescent labelling of TAD domains was adapted from[36] with some modifications. Briefly, cells were subjected to cell cycle arrest with 1 µg/ml aphidicolin (Cat.# A0781, Sigma) for 9 h, and subsequently allowed to recover and enter S-phase by incubation in fresh DMEM media for 20 min. Afterwards, cells were trypsinized, washed with PBS and resuspended in Resuspension Buffer R (Neon™ Transfection System 10 µL Kit, Cat.# MPK1025, Invitrogen) at a final density of $5.5 \times 10^6$ cells/ml. dUTP-AF647 (Cat.# NU-803-XX-AF647-L, Jena Biosciences) was added at a final concentration of 60 µM and cells were electroporated with the following pulse parameters: 1 pulse, 1100 V, 20 ms with the Neon transfection system (Invitrogen). Labeled cells were grown in complete DMEM media, passaged after 1 day and seeded in microslide chambers 3 days after transfection. For non-denaturing FISH experiments, 40 µM BrdU/C was added at the seeding step.

### SIM image acquisition and analysis

RPE-1 cells pulse-labeled with fluorescent nucleotides were incubated with 1 µg/mL Hoechst 33258 (Cat.# 14530, Sigma) in 1xPBS for 30 min. To avoid photobleaching, samples were imaged in an oxygen scavenger buffer system (100 mM tris, 50 mM NaCl, 0.5 mg/mL glucose oxidase, 40 µg/mL catalase, 10% glucose, 1 mM trolox for imaging before FISH, with tris and NaCl replaced with 2XSSC for imaging after

FISH). SIM images were acquired on a Zeiss Elyra 7 with a 63 × 1.4NA oil immersion objective using lattice SIM mode. Raw pixel size was 97 nm (1280 × 1280 pixels), z-spacing was 144 nm and camera (pco edge) exposure time was 50 ms. DNA (Hoechst, 405 nm laser) and replication domain (AF647, 642 nm laser) channels were acquired sequentially. Raw SIM images were processed in Zen Black 3.0 using standard settings except sharpening was set to "weak", resulting in images with twofold increase in pixels in each dimension. To compare images acquired before and after steps in the FISH protocol a global 3D translational drift correction was first applied using SciPy[66]. Single nuclei were detected and segmented using Cellpose[67], and the central plane of the nucleus was identified as the plane with the highest integrated intensity, and the corresponding plane was found in the comparison image as the plane with highest Pearson correlation. 2D slices of the central plane and the central plane + 8 slices were used for comparison. Maximum projections were used for the replication domains, and individual clusters of replication domains were segmented by Otsu thresholding. Images were cropped to the detected nuclei or replication domains, and a second pixel-level drift correction was applied to both rotation and translation (imreg-dft package). Finally, a subpixel translational registration was applied, before the pixel-wise Pearson correlation coefficient between the aligned images was calculated. The values for the two selected nuclear planes were averaged.

### Automated fluidics setup

A custom fluidics setup was built using a commercial 3-axis 30 × 18 cm GRBL controlled CNC stage. A syringe needle was mounted in place of the CNC drill head and connected using 1 mm i.d. PEEK and silicone tubing (VWR) to a male Luer adapter, which connected to the sample microslide. The microslide output tubing was connected to a mp6-liquid piezo micropump (Bartel's mikroteknik), connected to an MP-X controller (Bartel's mikroteknik) running at 130 Hz and 180 Vpp when dispensing, giving a flow rate of ~3.5 mL/min, or a CPP1 peristaltic micropump (Jobst Technologies, Freiburg, Germany) running at full speed giving a flow rate of ~1 mL/min. Probe, wash and imaging buffers were kept in 96 well or 4 well deep plates covered in parafilm (Cat.# P7793, Sigma) and placed on the CNC bed, and the appropriate solution was chosen by moving the syringe needle into the liquid through the parafilm. The automated control software was written in Python and was run on a PC connected to the MP-X controller/Jobst Pump Driver and GRBL CNC board. Detailed build and running instructions are described in the accompanying code repository (see code availability).

### 3D DNA trace acquisition

Cells on microslides were prepared for FISH as above, omitting secondary hybridization. 100 nm fluorescent fiducial beads (infrared Cat.# F8799, ThermoFisher or red Cat.# F8801, ThermoFisher) were diluted 1:20 000 in 2XSSC and applied to the sample for 10 min before washing in 2XSSC. 12 mer Atto565-labeled or Atto647N-labeled secondary imaging oligos were diluted to 20 nM in hybridization buffer (5% ethylene carbonate (EC, Cat.# E26258, Sigma) in 2XSSC/0.2% tween). Washing buffer was 10% FA in 2XSSC/0.2% tween, imaging buffer was 2XSSC with 1 µg/mL Hoechst 33258 and probe stripping buffer was 30% FA in 2XSSC/0.2% tween. Probes were hybridized for 3 min and washed for 1 min before switching to imaging buffer and imaging. After imaging, probes were stripped for 2 min, then wash buffer was flowed through for 1 min before the next round of probe hybridization. Imaging was performed on a Zeiss Elyra 7 widefield system using a 63 × 1.46NA oil immersion objective, a pco edge sCMOS camera and HILO mode, or a Nikon TI-E2 with a Lumencor Spectra III light engine, a 60 × 1.4 NA oil immersion objective and an Orca Fusion CMOS camera and widefield mode. 3D stacks of Hoechst (405 nm laser), FISH probes (561 or 642 nm excitation) and fiducial beads (561

or 642 nm excitation) were acquired as sequential frames at each z-position. The pixel size was 97 nm (Zeiss Elyra) or 111 nm (Nikon Ti-E2), image size 1280 × 1280 (Zeiss Elyra) or 2304 × 2304 (Nikon Ti-E2) and z-spacing 150 or 200 nm, and the camera exposure time was 50 ms (Zeiss Elyra) or 100 ms (Nikon Ti-E2). 15–60 fields of view were acquired per round of hybridization. Total acquisition time for a typical experiment with 20–30 fields of view and 14–40 hybridizations was 4 – 20 h. The Zeiss microscope was controlled using Zen 3.0 software (Carl Zeiss), with the MyPic macro[68,69], while the Nikon Ti-E2 was controlled using NIS Elements 5.2.02 (Nikon). For both systems custom Python software for automation and synchronization with the microfluidic software was used (see code availability). In experiments with multiple mixed ViaFluor labeled cell lines, the entire channel was imaged before FISH with 405 nm (VF405) and 475 nm (VF488) excitation and a 20 × 0.8 air objective, relabeled with DAPI (100 ng/mL in PBS for 15 min) and imaged again using the same settings before proceeding to DNA tracing as above.

## DNA trace fitting

Custom Python software was developed to extract, fit and analyse DNA traces from the sequential FISH images (see code availability). Raw Zeiss CZI or Nikon ND2 images converted to the ZARR file format and deconvolved with a theoretical microscope-appropriate point spread function (10–30 iterations) or an experimental point spread function (60 iterations) with a Richardson-Lucy iterative algorithm. Subpixel drift correction was calculated for all hybridization rounds by upsampled cross-correlation or Gaussian centroid fitting of 50–200 segmented fiducial beads and averaged. Regions of interest (ROIs) with FISH signal were automatically identified based on an empirical threshold in frames with imagers targeting most or all positions in a region for brighter signal. For weaker signal, enhancement of spot-like signals for detection using a difference of gaussian filter. When multiplexing tracing of multiple regions in the same cell, different regions were simultaneously traced and individual regions were identified by hybridization with region-specific probes, and ROIs were detected for each region in the respective frames. Nuclei were detected using Cellpose and only ROIs inside nuclei were included. Detected ROIs were quality controlled by manual spot-checks. For decoding VF405 and VF488-labeled cell lines (see Supplementary Fig. S8b), nuclei images were cross-correlated with the low magnification images acquired before FISH treatment, then VF405 and VF488 intensities in 5-pixel dilated nuclear masks were manually selected on a scatterplot to decode the cell line identity.

The ROIs with FISH signals were cropped and the 3D image from each hybridization was fit to a 3D Gaussian function by least squares minimization, with the centre defining the coordinate in the corresponding DNA trace.

## DNA trace analysis

3D DNA traces were analysed using custom Python software (see code availability). All fits were quality controlled by empirical thresholds of signal to background, fit standard deviation and distance to the multiple-probe/regional probe signal to avoid fitting of spurious background signals. Due to the non-denaturing FISH procedure, no traces from sister chromatids are generated and so did not require additional filtering. DNA traces were filtered for minimum percentage of quality-controlled fit loci of 67% for consensus representations, distance/contact maps and pairwise similarity measures, and 80% for contact counting.

Pairwise distance maps and contacts maps were calculated by directly calculating median pairwise distances from the measured 3D coordinates, and the frequencies of those distances less than a cut-off (set to 100 nm). Consensus representations were generated by a general Procrustes alignment using an iterative pairwise alignment to an initially randomly chosen template. Pairwise alignments were performed by 3D rigid alignment using singular value decomposition (SVD) without scaling to minimise the error (RMSE). A quadratic spline interpolation was used to connect the fit loci coordinates for visualization. 3D traces were visualized with mayavi. Consensus representations of multiple traces were overlaid with the per-loci standard deviation of the single traces compared to the Procrustes consensus.

Seriation and clustering of traces was performed by calculating all pairwise distances of each trace, taking the reciprocal of each value to emphasise shorter distances, and then calculating the pairwise difference matrix between all traces with the Pearson distance ($\sqrt{1 - PCC}$) as the distance metric. Seriation was performed by ordering the traces according to the second largest eigenvector of the difference matrix (Fiedler vector)[40] and splitting the resulting ordered traces into six groups of equal size. Clustering of traces was done after UMAP dimensionality reduction (umap-learn) with n_neighbor = 20 and min_dist = 0 by Variational Bayesian estimation of a Gaussian mixture (scikit-learn) with n_components = 10 and other parameters at default values. To rule out that the obtained clusters could arise by chance, we compared average intra-cluster distances (0.697 ± 0.11) to intra-cluster distances after randomly assigning traces to clusters of corresponding sizes (0.92 ± 0.01), which validated that real clusters grouped significantly more similar traces than by random chance (Supplementary Fig. S8c).

## Simulations

Ideal random coils were generated using 3D random walkers, where each step was in a random direction and had a length drawn from a standard normal distribution. Rouse polymer models with and without loop extrusion were established using Brownian dynamics in the Euler scheme, as described in detail in Supplementary Note 1. Simulation parameters are provided in Supplementary Table 1.

## Visualization and annotation of sequencing data

HiC maps and CHIP-seq data were visualized using pyGenomeTracks. CTCF motif orientation was annotated on the CTCF CHIP-seq data using fimo v5.3.0 with the MA0139.1 motif and a p-value threshold of 0.01. The motif overlapping the CTCF Chip-seq peak with the lowest fimo p-value was used.

## Statistics and Reproducibility

Statistical analyses were performed using scipy 1.14.1. Appropriate tests were applied as indicated in the figure legends. Sample sizes of 50–500 single cells per replicate were based on pilot experiments to achieve stable population averages (Pearson's $r$ = 0.98–0.99 across replicates). All data that passed our predefined quality-control (QC) metrics for spot fitting and trace length were included in the analyses, and lower-quality traces were filtered out as described in Methods. Each experiment was performed in two to three independent biological replicates. For each replicate, cells were imaged in 10–40 randomly chosen fields of view and all QC-passed cells were analyzed. No predefined group-allocation, randomization or investigator blinding was used as data processing and analysis were fully automated and applied uniformly to all samples.

## Reporting summary

Further information on research design is available in the Nature Portfolio Reporting Summary linked to this article.

# Data availability

The DNA tracing data generated in this study have been deposited in figshare under accession code 27613239 [https://doi.org/10.6084/m9.figshare.27613239]. The RPE-1 HiC data used in this study are available in the GEO database under accession code GSE71831. The HeLa HiC data used in this study are available in the GEO database under accession code GSE63525. The HeLa CTCF ChIP-seq data used in this

study are available in the ENCODE database under accession code ENCSR000AOA. The HeLa RAD21 ChIP-seq data used in this study are available in the ENCODE database under accession code ENCSR000EDE. Source data are provided with this paper.

## Code availability

Custom code for automated fluidics control is available online at https://git.embl.de/grp-ellenberg/tracebot (release v1.0). Custom code of image processing and analysis, data analysis and Rouse polymer simulation code is available online at https://git.embl.de/grp-ellenberg/looptrace (release v0.3).

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

## Acknowledgements

We thank Merle Hantsche-Grininger for developing the electroporation protocol for replication domain labelling and critical reading of the manuscript; Franziska Kundel, Jonas Ries and Virginie Uhlmann for discussions; Magda Bienko and Jill Brown for sharing of FISH protocols; the EMBL Advanced Light Microscopy Facility (ALMF) for microscope support; Christian Tischer and Jean-Karim Hériché for help with data/metadata handling, the electronic workshop and mechanical workshop at EMBL for assistance with the fluidics setup and the EMBL HPC cluster for computational infrastructure. This work was supported by grants from the National Institutes of Health Common Fund 4D Nucleome Program (Grant U01 EB021223 / U01 DA047728) to J.E. and E.B., and The Paul G. Allen Frontiers Group through the Allen Distinguished Investigator Program to J.E. and R.J (# 12359), as well as by the European Molecular Biology Laboratory (K.S.B., Ø.Ø.-F., N.M, S.A., J.E.). K.S.B. was also supported by the Alexander von Humboldt foundation. Work in the laboratory of J.-M.P. has received funding from Boehringer Ingelheim, the Austrian Research Promotion Agency (Headquarter grant FFG-852936), the European Research Council (ERC) under the European Union's Horizon 2020 research and innovation programme (grant agreements No 693949 and No 101020558), the Human Frontier Science Program (grant RGP0057/2018) and the Vienna Science and Technology Fund (grant LS19-029).

## Author contributions

Conceptualization: J.E. and E.B. Methodology: Ø.Ø.F, K.S.B., N.R.M., C.B., F.S., T.W. Software: K.S.B., C.B. Validation: K.S.B., N.R.M., T.W. Formal analysis: K.S.B. Investigation: Ø.Ø.F., K.S.B., N.R.M. Data Curation: K.S.B., C.B. Writing – original draft preparation: Ø.Ø.F., K.S.B., J.E. Writing – review and editing: All authors. Visualization: K.S.B. Supervision: S.A., J.M.P., E.B., R.J., J.E. Funding acquisition: K.S.B., S.A., J.M.P., E.B., R.J., J.E.

## Funding

## Competing interests

The authors declare no competing interests.

## Additional information

**Peer review information** \Nature Communications thanks the anonymous reviewers for their contribution to the peer review of this work. A peer review file is available.

