## [Peer Review file · Nature Communications]

Nanoscale 3D DNA tracing in non-denatured cells resolves the Cohesin-dependent loop architecture of the genome in situ

Corresponding Author: Dr Jan Ellenberg

Version 0:

Reviewer comments:

Reviewer #1

(Remarks to the Author)

The authors have resubmitted a revised manuscript and transferred from Nature Cell Biology. While I had some concerns, I strongly supported publication at Nature Cell Biology, and I now even more strongly support publication.

The authors have for the most part addressed my concerns. My concerns were mainly related to overclaiming about detecting loop extrusion in situ, and I think the revised manuscript is balanced.

I also think the technical concerns I had, have been addressed. All my remaining comments are very minor things (see below) to increase clarity for the reader.

If I may, I would also like to comment on the other reviews. Reviewer 2 and 3 make what in my view constitutes non-scientific arguments that use “appeal to authority” instead of data assessment. E.g. R3 mentions that “labs of Wu, Zhuang, Boettiger, Cai, Cavalli” did not report distortions. Of course, those labs would have an interest in not overreporting such distortions (though I very much do not want to impugn their intentions). However, the regular FISH protocol involves heating fixed cells in formamide to typically 80-95C – it would be shocking if heating cells to >80C in formamide did not move and rearrange chromatin by tens if not hundreds of nanometer. Furthermore, the authors have very convincingly shown that there are artifacts, which is hardly surprising. The authors also make a very convincing case that the resolution of SIM is required to detect the distortions. I think the data presented by the authors on this is extremely solid.

Therefore, I would like to state that I found some of this pushback unfair to the authors and I think their paper should be published as soon as possible to report these timeline and important results.

SMALL THINGS

I am very confused by Fig 4c. It contains two plots, chr17 and chr6. On the left, the exponent is ~0.3-0.35 for all except dRAD21. But on the right it is ~0.5 for all except dCTCF. So the same condition has 0.3 on one side but 0.5 on the other side. Am I missing something? This is self-contradictory?

Lines 304-308, the concept of loop stacking has been popularized by REF 25, but it was well-explained and predicted by e.g. Fudenberg 2016 and the early loop extrusion papers, so I think it would be good to mention this since the concept was long-known before REF 25.

Line 401 refers to Fig 4e, but Fig 4 only had a-d panels?

Around lines 350-400 the authors frequently make statements such as “32% of Cohesin-specific loops between the two convergent CTCF sites”, but it is not clear to me how this relates to the data presented in the figures. The figures tend to show histograms and example traces – can you please make clear how these numbers appear and where they come from in the figures?

(Remarks on code availability)

Reviewer #2

(Remarks to the Author)

The authors have made very important, and critical improvements to the previous version of the manuscript by revising the text, and by adding additional analysis and experiments. However, the manuscript has several significant issues that would need to be addressed, specifically the lack of supporting evidence for some of their key conclusions, the interpretation of some of the results/analysis, the revision of some of the figures. See below for details.

Major issues

Fig. S2c is very interesting as it shows that distances by DNA-FISH are larger than for raser-FISH at all genomic scales. This by itself shows that the difference between these methods does not arise at the 50-100nm scale. Also, this shows that a simple pre-factor in the distances may be able to correct the values in pairwise distances from DNA-FISH to fit those obtained from raser-FISH. Can the authors show a pairwise distance map normalized so that the physical-to-genomic curves in S2c overlap? This would be a very important control for the community, as raser-FISH can only be applied to certain model systems where BrdU can be incorporated.

The manuscript makes statements that are not supported by data. For instance, when describing their method, they state that it ‘...preserves the nanoscale structure of the genome in a near-native state.’ It is reasonable to expect that raser-FISH will be milder than DNA-FISH, and they clearly demonstrate that the variance in their pairwise distance distributions is lower for the former. However, they never show in the manuscript that their sequential method preserves the nanoscale structure near the native state. Experiments would be required to support this statement.

I remain skeptical about the meaning of a 'consensus' structure (e.g., Figs. 3c-f). A polymer structure is by nature dynamic, and there is no sense in discussing the 'structure of a polymer' such as chromatin. In addition to being incorrect, it will also be misleading to biologists without the right background to realize that this is not a structure. Therefore, I strongly discourage the authors from calling this a structure or a consensus trace, but perhaps suggest using a term such as 'representation.'

From Figures 3c-f, the authors conclude that spectral seriation clustering is a useful tool to identify diverse folds. However, I don't see the experimental proof that this is the case. For this, they could simulate mixes of conformations (extruded, no loop extrusion, etc.) to validate that these ground-truth structures are actually recovered by their method.

I do not follow the meaning of the sentence in lines 276-277, nor the interpretation of the plots in Fig. S5g. Also, why does there not seem to be a difference between WT and dCTCF in Fig. S5g if these plots represent loop stacking between CTCF sites?

I perhaps misunderstand the data presented, but it seems from the inspection of Figs. 6e (particularly the PWD maps) that the simulations do not produce the same pairwise distance maps as experiments. If this is the case, why do the authors insist that their simulations accurately predict the 3D folding of genomic DNA up to 2Mb? Such a strong statement warrants more supporting evidence. Displaying a single conformation from a simulation and an experiment is not enough, as this does not show that the ensemble of structures from simulations faithfully reproduces the ensemble of structures from experiments.

This lack of validation of the model puts into question their claim that they can now predict the key parameters of individual loops (lines 392).

Fig. 2e: Values for negative distance values in this figure seem artifactual. How are these distributions calculated? Similar comments apply to panels in Fig. S2d. These plots do not allow the reader to clearly see the differences in the maxima of the distributions from raser-FISH and by DNA-FISH. Particularly for probes that are located at short distances (~10kb). From the plots mentioned above, it seems both methods provide distances that are around 100nm?

Minor issues

Fig. 2f: Why plot all the distributions together if these correspond to different genomic distances and therefore their mean makes little sense? In this case, the authors can just perform a scatter plot of the median distance from raser-FISH against that from DNA-FISH (as they did when comparing HiC and raser/DNA-FISH).

Fig. S2b: the authors call the Hi-C counts 'ligation frequency.' While the Hi-C counts depend on the ligation frequency, this is not the only factor affecting this measurement.

Line 141: "was due to structural perturbation" instead of "was due structural perturbation"

Lines 228-233, Fig. 3i, lines 1057-1064: Unsure if the statement in the text is justified based on the figure and the values for the radius of gyration and the elongation ratio provided in the caption. Also, the caption numbering is outdated -- should it be i instead of j, and h instead of i?

Lines 260-266, Fig. 4c: Surprising that they observe an $R(s)$ scaling exponent corresponding to an ideal chain on the right, and that the depletion of CTCF results in a lower scaling exponent. What could be the reason? Has this ever been seen before in the literature? They must discuss this somewhere because this does not seem to be a result that agrees with the literature.

Lines 297-301, Fig. 5b: The quantification of the long-range contact between CTCF sites that is in Fig. S6e could be included in the main panel in a column next to Fig. 5b

Line 843: What is the source or rationale for the empirical conversion factor $24.7 \text{ nm kb}^{-1/2}$?

(Remarks on code availability)

Reviewer #4

(Remarks to the Author)

(Remarks on code availability)

Code is available, but authors must create a versioned release of the code. This version must be referred to in the manuscript, in order to ensure reproducibility.

Version 1:

Reviewer comments:

Reviewer #1

(Remarks to the Author)

This paper is a very important paper for the field. Figure 2 is extremely useful because it clearly shows the issues with denaturing FISH. Beyond technical aspects, the biological insights are significant. I very strongly encourage publication of the paper - in its present form without any further modification - as soon as possible.

(Remarks on code availability)

code is good

Reviewer #2

(Remarks to the Author)

The authors have done an excellent job at addressing my concerns

(Remarks on code availability)

Reviewer #4

(Remarks to the Author)

The authors have responded to my comment satisfactorily

(Remarks on code availability)

The code is available via git.embl.de/grp-ellenberg/ and is now versioned and released properly, with versions mentioned in the 'code availability' section.

Reviewer #1 (Remarks to the Author):

The authors have resubmitted a revised manuscript and transferred from Nature Cell Biology. While I had some concerns, I strongly supported publication at Nature Cell Biology, and I now even more strongly support publication.

The authors have for the most part addressed my concerns. My concerns were mainly related to overclaiming about detecting loop extrusion in situ, and I think the revised manuscript is balanced.

I also think the technical concerns I had, have been addressed. All my remaining comments are very minor things (see below) to increase clarity for the reader.

If I may, I would also like to comment on the other reviews. Reviewer 2 and 3 make what in my view constitutes non-scientific arguments that use “appeal to authority” instead of data assessment. E.g. R3 mentions that “labs of Wu, Zhuang, Boettiger, Cai, Cavalli” did not report distortions. Of course, those labs would have an interest in not overreporting such distortions (though I very much do not want to impugn their intentions). However, the regular FISH protocol involves heating fixed cells in formamide to typically 80-95C – it would be shocking if heating cells to >80C in formamide did not move and rearrange chromatin by tens if not hundreds of nanometer. Furthermore, the authors have very convincingly shown that there are artifacts, which is hardly surprising. The authors also make a very convincing case that the resolution of SIM is required to detect the distortions. I think the data presented by the authors on this is extremely solid.

Therefore, I would like to state that I found some of this pushback unfair to the authors and I think their paper should be published as soon as possible to report these timeline and important results.

We thank the reviewer for these positive comments on our work.

SMALL THINGS

I am very confused by Fig 4c. It contains two plots, chr17 and chr6. On the left, the exponent is ~0.3-0.35 for all except dRAD21. But on the right it is ~0.5 for all except dCTCF. So the same condition has 0.3 on one side but 0.5 on the other side. Am I missing something? This is self-contradictory?

We thank the reviewer for highlighting this point. We chose to display these two regions in the main figure precisely because they showed interesting differences in the scaling upon CTCF and Cohesin depletion, while two other regions on Chr10 and 12 (Supplementary Figure 5c) showed similar scaling features as the region on Chr17. We have now highlighted this point explicitly in the text as follows.

“However, regional differences were observed. For example, the region on Chr6 was already unconstrained in WT cells and did not further decompact upon Cohesin depletion.”

(...)

“Interestingly, CTCF depletion compacted the region on Chr6 to match the other regions (Fig. S5b), suggesting that in this region CTCF prevents Cohesin from extruding longer loops, reminiscent of how CTCF acts to prevent looping across TAD boundaries [5, 42].”

Lines 304-308, the concept of loop stacking has been popularized by REF 25, but it was well-explained and predicted by e.g. Fudenberg 2016 and the early loop extrusion papers, so I think it would be good to mention this since the concept was long-known before REF 25.

We agree and have added additional references to this section.

Line 401 refers to Fig 4e, but Fig 4 only had a-d panels?

We thank the reviewer for noticing this error and have corrected the reference to Fig. 3h/Fig. S4d.

Around lines 350-400 the authors frequently make statements such as “32% of Cohesin-specific loops between the two convergent CTCF sites”, but it is not clear to me how this relates to the data presented in the figures. The figures tend to show histograms and example traces – can you please make clear how these numbers appear and where they come from in the figures?

We thank the reviewer for pointing this out and have now clarified that these values refer to loops predicted by our loop extrusion simulations shown in Fig. 6b.

Reviewer #2 (Remarks to the Author):

The authors have made very important, and critical improvements to the previous version of the manuscript by revising the text, and by adding additional analysis and experiments. However, the manuscript has several significant issues that would need to

be addressed, specifically the lack of supporting evidence for some of their key conclusions, the interpretation of some of the results/analysis, the revision of some of the figures. See below for details.

We thank the reviewer for appreciating the additional data provided in our revised manuscript.

Major issues

Fig. S2c is very interesting as it shows that distances by DNA-FISH are larger than for raser-FISH at all genomic scales. This by itself shows that the difference between these methods does not arise at the 50-100nm scale. Also, this shows that a simple pre-factor in the distances may be able to correct the values in pairwise distances from DNA-FISH to fit those obtained from raser-FISH. Can the authors show a pairwise distance map normalized so that the physical-to-genomic curves in S2c overlap? This would be a very important control for the community, as raser-FISH can only be applied to certain model systems where BrdU can be incorporated.

We thank the reviewer for these comments. We would like to highlight that our results comparing denaturing and RASER-FISH shown qualitatively in Fig. 2b and by increased spread of point-to-point distances shown in Fig. 2e and f indicates that the distortions introduced by acid/heat denaturation are not uniform but rather cause both local accumulation and spreading of chromatin. We appreciate that BrdU cannot be used in many systems and a way to calibrate data across FISH protocols would be useful. Nevertheless, we believe it would be misleading to rescale average pairwise distances as suggested, as this would imply a more uniform profile of the distortions than is supported by the data.

The manuscript makes statements that are not supported by data. For instance, when describing their method, they state that it '...preserves the nanoscale structure of the genome in a near-native state.' It is reasonable to expect that raser-FISH will be milder than DNA-FISH, and they clearly demonstrate that the variance in their pairwise distance distributions is lower for the former. However, they never show in the manuscript that their sequential method preserves the nanoscale structure near the native state. Experiments would be required to support this statement.

This statement refers to the SIM data shown in Figure S3, which shows fewer distortions using RASER-FISH than denaturing DNA-FISH compared to samples of TAD-scale replication domains at ~120 nm resolution. We acknowledge that this approach still has

limitations, as highlighted in the discussion section. Although we welcome further validations, it is experimentally challenging to measure high-resolution chromatin structure in unperturbed cells. We nevertheless acknowledge that “near-native” could be interpreted in different ways and have replaced this term in the manuscript to better highlight that structural preservation by RASER-FISH is improved compared to denaturing protocols.

I remain skeptical about the meaning of a 'consensus' structure (e.g., Figs. 3c-f). A polymer structure is by nature dynamic, and there is no sense in discussing the 'structure of a polymer' such as chromatin. In addition to being incorrect, it will also be misleading to biologists without the right background to realize that this is not a structure. Therefore, I strongly discourage the authors from calling this a structure or a consensus trace, but perhaps suggest using a term such as 'representation.'

We agree that finding good representations of structures that encompass the inherent structural variability of chromatin remains challenging. We appreciate the suggested term “consensus representations” and have implemented this term throughout the manuscript.

From Figures 3c-f, the authors conclude that spectral seriation clustering is a useful tool to identify diverse folds. However, I don't see the experimental proof that this is the case. For this, they could simulate mixes of conformations (extruded, no loop extrusion, etc.) to validate that these ground-truth structures are actually recovered by their method.

In Fig. 6b spectral seriation results from polymer loop extrusion simulations are shown, which include both unextruded, partially extruded and fully extruded loops. This figure indicates that states enriched for these ground-truth conformations are enriched in the different spectral seriation groups.

I do not follow the meaning of the sentence in lines 276-277, nor the interpretation of the plots in Fig. S5g. Also, why does there not seem to be a difference between WT and dCTCF in Fig. S5g if these plots represent loop stacking between CTCF sites?

We have clarified the phrasing of this section, and added a comment regarding the similar proportion of stacked loops also without CTCF as follows:

“At genomic distances of 200-300 kb, around half of the contacts below 100 nm were formed by “stacking” of smaller loops (defined by two loops sharing an anchor to form a 3-way contact) (Fig. S5g) [9, 18, 25]. Interestingly, CTCF-depleted cells had a similar proportion of stacked loops, suggesting that at this genomic scale there is a sufficient density of Cohesin to create stacked loops also in the absence of CTCF.”

I perhaps misunderstand the data presented, but it seems from the inspection of Figs. 6e (particularly the PWD maps) that the simulations do not produce the same pairwise distance maps as experiments. If this is the case, why do the authors insist that their simulations accurately predict the 3D folding of genomic DNA up to 2Mb? Such a strong statement warrants more supporting evidence. Displaying a single conformation from a simulation and an experiment is not enough, as this does not show that the ensemble of structures from simulations faithfully reproduces the ensemble of structures from experiments.

This lack of validation of the model puts into question their claim that they can now predict the key parameters of individual loops (lines 392).

We acknowledge that our simulations do not perfectly match the tracing data of all genomic regions. In our simulations, we assume that all dynamically bound CTCFs have an equal chance of stalling a passing Cohesin, and that the simulated occupancy of CTCFs bound at each site is determined by Chip-seq signal strength. Recent studies suggest that the interactions of CTCF and Cohesins may be more complex (Huang et.al. *Nature Genetics*, 2021; Tsang et.al. *NAR* 2024; Smaruj et.al. *Plos Biology*, 2025), which could explain the differences between the simulated and measured pairwise distance maps.

Nevertheless, in Fig. 6c we show that the pairwise similarity within the ensemble of ~100 kb looping structures matches well to experimental data, and the same is true for the simulated seriation groups (Fig. 6b), validating that we can simulate ensembles of loop-scale structures matching experimental measurements. In addition, the scaling plots Fig. S7b highlight that the overall pairwise distance scaling is well matched between experiments and simulations.

We have expanded our discussion to better emphasize the limitations of the simulations, and rephrased selected statements in the results. The discussion now contains the following paragraph highlighting these limitations:

“The increased CTCF abundance suggested by the simulations may be due to the assumption that all dynamically bound CTCFs have an equal chance of stalling a passing Cohesin, and that the occupancy of CTCF scales with ChIP-seq signal strength. Recent studies suggest that the looping frequency of specific loops and thus the resulting pairwise distance maps of the genomic regions may be more complex [56-58]. However, the mechanistic basis for this discrepancy remains unclear, and before making more complex assumptions in our simulations, such as constrained diffusion of CTCF and CTCF clustering [55] and/or stabilization of chromatin-bound CTCF by Cohesin [54], further investigations would be needed.”

Fig. 2e: Values for negative distance values in this figure seem artificial. How are these distributions calculated? Similar comments apply to panels in Fig. S2d. These plots do not allow the reader to clearly see the differences in the maxima of the distributions from raser-FISH and by DNA-FISH. Particularly for probes that are located at short distances (~10kb). From the plots mentioned above, it seems both methods provide distances that are around 100nm?

The apparent presence of negative values is due to the use of KDE smoothing in the representation of the data, as there are indeed no negative values in the dataset. We have replaced the previous plots with histograms in Fig. 2e and Supp. Fig. S2d to better represent these distributions of pairwise distance, which emphasize that an increased spread in the pairwise distance distribution is an important effect of denaturing FISH.

Minor issues

Fig. 2f: Why plot all the distributions together if these correspond to different genomic distances and therefore their mean makes little sense? In this case, the authors can just perform a scatter plot of the median distance from raser-FISH against that from DNA-FISH (as they did when comparing HiC and raser/DNA-FISH).

We agree that grouping pairwise distances and their standard deviations for all genomic distances together obscures aspects of the data. This representation was chosen to highlight the consistency of the findings across different genomic regions. We have now removed Fig. 2f as a more accurate representation of the median pairwise distances across genomic distances is displayed in Fig. 2d for one region and Supplementary Fig. S2c for the other regions.

Fig. S2b: the authors call the Hi-C counts 'ligation frequency.' While the Hi-C counts depend on the ligation frequency, this is not the only factor affecting this measurement.

We thank the reviewer for pointing this out and have changed the HiC annotation to the commonly used term "contact frequency".

Line 141: "was due to structural perturbation" instead of "was due structural perturbation"

Thank you, corrected.

Lines 228-233, Fig. 3i, lines 1057-1064: Unsure if the statement in the text is justified based on the figure and the values for the radius of gyration and the elongation ratio

provided in the caption. Also, the caption numbering is outdated -- should it be i instead of j, and h instead of i?

We thank the reviewer for pointing out the error in caption lettering. We have provided more details in the figure legend on the statistical test that underlies the p-values plotted in the figure that shows that although there are overlapping standard deviations as highlighted in the figure caption, the indicated difference in median values are significant which supports the statement in the text.

Lines 260-266, Fig. 4c: Surprising that they observe an $R(s)$ scaling exponent corresponding to an ideal chain on the right, and that the depletion of CTCF results in a lower scaling exponent. What could be the reason? Has this ever been seen before in the literature? They must discuss this somewhere because this does not seem to be a result that agrees with the literature.

We thank the reviewer for highlighting this interesting point about the scaling behavior of the region. We have added the following sentence to the relevant paragraph to provide our interpretation of these observations:

“Interestingly, CTCF depletion compacted the region on Chr6 to match the other regions (Fig. S5b), suggesting that in this region CTCF prevents Cohesin from extruding longer loops, reminiscent of how CTCF acts to prevent looping across TAD boundaries [5, 42].”

Lines 297-301, Fig. 5b: The quantification of the long-range contact between CTCF sites that is in Fig. S6e could be included in the main panel in a column next to Fig. 5b

We thank the reviewer for the suggestion, and have now moved Fig. S6e to main figure 5.

Line 843: What is the source or rationale for the empirical conversion factor $24.7 \text{ nm kb}^{-1/2}$?

This factor was derived from earlier tracing data that was removed from the manuscript during the revisions. The scaling factor is no longer relevant as the application of these random coils is now only as a control of the seriation results, which in this case does not depend on the physical scaling. This has been revised in the methods section.

Reviewer #3 (declined to re-review)

Reviewer #4 (Remarks on code availability):

Code is available, but authors must create a versioned release of the code. This version must be referred to in the manuscript, in order to ensure reproducibility.

We have created versioned code release associated with our repository for downloading and have referenced this version in the manuscript.

Reviewer #1 (Remarks to the Author):

This paper is a very important paper for the field. Figure 2 is extremely useful because it clearly shows the issues with denaturing FISH. Beyond technical aspects, the biological insights are significant. I very strongly encourage publication of the paper - in its present form without any further modification - as soon as possible.

Reviewer #1 (Remarks on code availability):

code is good

Reviewer #2 (Remarks to the Author):

The authors have done an excellent job at addressing my concerns

Reviewer #4 (Remarks to the Author):

The authors have responded to my comment satisfactorily

Reviewer #4 (Remarks on code availability):

The code is available via git.embl.de/grp-ellenberg/ and is now versioned and released properly, with versions mentioned in the 'code availability' section.

We thank the reviewers for the positive assessment of our work, and have no further comments.